# Improving Self-supervised Learning with Automated Unsupervised Outlier Arbitration

**Yu Wang**[1]    **Jingyang Lin**[2]    **Jingjing Zou**[3]    **Yingwei Pan**[1]    **Ting Yao**[1]    **Tao Mei**[1]

[1] JD AI Research, Beijing, China
[2] Sun Yat-sen University, Guangzhou, China
[3] University of California, San Diego, USA

{feather1014, yung.linjy}@gmail.com, j2zou@ucsd.edu
{panyw.ustc, tingyao.ustc}@gmail.com, tmei@jd.com

## Abstract

Our work reveals a structured shortcoming of the existing mainstream self-supervised learning methods. Whereas self-supervised learning frameworks usually take the prevailing perfect instance level invariance hypothesis for granted, we carefully investigate the pitfalls behind. Particularly, we argue that the existing augmentation pipeline for generating multiple positive views naturally introduces out-of-distribution (OOD) samples that undermine the learning of the downstream tasks. Generating diverse positive augmentations on the input does not always pay off in benefiting downstream tasks. To overcome this inherent deficiency, we introduce a lightweight latent variable model UOTA, targeting the view sampling issue for self-supervised learning. UOTA adaptively searches for the most important sampling region to produce views, and provides viable choice for outlier-robust self-supervised learning approaches. Our method directly generalizes to many mainstream self-supervised learning approaches, regardless of the loss's nature contrastive or not. We empirically show UOTA's advantage over the state-of-the-art self-supervised paradigms with evident margin, which well justifies the existence of the OOD sample issue embedded in the existing approaches. Especially, we theoretically prove that the merits of the proposal boil down to guaranteed estimator variance and bias reduction. Code is available: https://github.com/ssl-codelab/uota.

## 1 Introduction

Self-supervised learning is an increasingly appealing direction in learning effective deep representations. In the regime of computer vision, natural language processing and machine learning tasks, multiple milestones under such self-supervised learning frameworks have been established [4, 5, 14, 12, 18, 32, 31, 33, 39, 43, 47, 48, 49]. The premise here is that self-supervised learning methods usually generate multiple views and assume one view be predictive of another.

So far, the most prevailing assumption is to force views from the same instance invariant in the feature space [7, 18, 33, 47]. While it is certainly natural to consider generating diverse augmentations to spread the feature distribution and force consistency in between views, we reveal a structured shortcoming of such popular augmentation pipeline: excessive distortions applied on original image would produce samples that deviate drastically in the semantics. We find these produced views have semantically deviated from the original instance and thus behave like out of distribution (OOD) samples. The model therefore fail to generalize in some certain parameter region due to the interference of the OOD samples.

We support our claim by empirically verifying the non-negligible detrimental impact of the OOD noise on downstream task performances. Particularly, we conduct extensive experiments and show that importance sampling technique and the associated analysis can help effectively differentiate the noises. The proposed new model suppresses the effect of noisy samples and efficiently improve the downstream task performance over a broad spectrum of state-of-the-art self-supervised learning

approaches. This indicates the evident failure of baseline model in some parameter space owing to the OOD noise. Our contribution in this paper are summarized as follows: 1. We present the first formal analysis of OOD issue under the currently prevailing self-supervised learning framework. 2. We propose a lightweight latent variable model UOTA that is able to differentiate noise from original instance's semantics. The new model does not introduce extra computational complexity whereas it can efficiently suppress the influence of OOD samples. In contrast to existing SSL paradigms, the proposal method is able to automatically balance the bias-variance trade-off in the mean squared error (MSE) of the deep estimator: We do desire large data variance through strong augmentations, on the other hand, we should be careful not to introduce extra estimator bias through extremely large data distortions. We argue that in the context of self-supervised learning, the hidden OOD samples and such bias-variance trade-off have a clear and non-negligible impact that should be attended to.

## 2 Related Work

**Self-supervised learning.** Without access to any data annotations, self-supervised learning (SSL) approaches impose additional constraints or hypothesis among distinct views of the original input data [2, 7, 13, 14, 32, 33, 39, 41, 44, 52]. Recently, mainstream approaches require that different views out of the same instance should be predicative of each others. Such line of research can be roughly classified into several sub-directions. The first family is inspired from the seminal NCE approach [16] and capitalizes on contrastive loss [5, 7, 17, 18, 24, 33, 38, 42]. These approaches usually rely on the construction of both the positive and negative views of a single instance. In these work, positive samples usually are augmented views of the same reference instance; Negative samples are defined as any views from a distinct instance other than that reference instance. Contrastive loss is then expected to learn useful semantics when forced to distinguish between positive and negative views on each instance. The second line of research further simplify the contrastive assumption, which only imposes invariant constraints between paired positive samples [9, 15, 46, 47] in the absence of the negative samples.

**Bias correction for contrastive learning**. Existing literature prioritize the sampling bias issue for negative samples, while positive samples are usually considered clean. In [10], debiased contrastive learning corrects the sampling bias in the negative samples in order to reduce the effect of false negatives. In [24, 37], harder negatives are created for training in order to improve the learning. The work in [30] is the only work we are aware of, that touches the noise issue in positive pairs. But [30] is motivated exclusively for observed caption-frames misalignments issues in narrated videos, and the loss is tailored for applications that are based on contrastive loss only. In contrast, we reveal that OOD noise actually exists in a broader coverage of self-supervised learning frameworks and is influencing the downstream tasks with a non-negligible strength.

**OOD detection and noisy label problem.** Outlier detection is an important problem in machine learning. A popular and intuitive framework for detecting OOD inputs is to train a generative model on training data and use that to detect OOD inputs at test time. For instance, in [34], a likelihood ratio method is proposed to correct for the background statistics between inliers and outliers. In the meanwhile, for supervised learning tasks, robust training in the presence of noisy labels is also a practical and challenging issue. In [50], a generalized cross entropy loss is used to mitigate the label noise issue. In [23], Mentor-net capitalizes on tailored network architecture to down-weight the samples likely to be associated with noisy labels. All of these work were targeting conventional supervised learning approaches, whereas the noise under the SSL framework is rarely explored.

**Comparison to existing works.** We unveil that the intrinsic positive sample noises exist broadly behind various mainstream SSL frameworks. To both justify and to mitigate the issue, we propose a latent variable framework that is capable improving the SSL optimization via automatic Unsupervised OuTlier Arbitration (UOTA). We are also aware of a contemporary work in [51] that attempts to change the sampling distribution via assigning sample weights to each positive sample. But motivation in [51] is completely opposite to ours, as they oversample the region whenever a positive sample turns out to be remote to its query, while they still assume a perfect invariance on positive keys as critical supporting prerequisite. In [26], a Mahalanobis distance-based confidence score is defined to modify sampling distribution for supervised learning tasks. Note our work differentiates from [26] in many aspects: we analyze the OOD issue particularly for unsupervised SSL frameworks, whereas [26] strongly hinges on data labels to compute confidence. We model the associated sampling ratio as a latent variable model which is exclusively suitable for the SSL problem, and is supported by rigorous analysis. Our model also naturally jettisons the need of any threshold as used in [26].

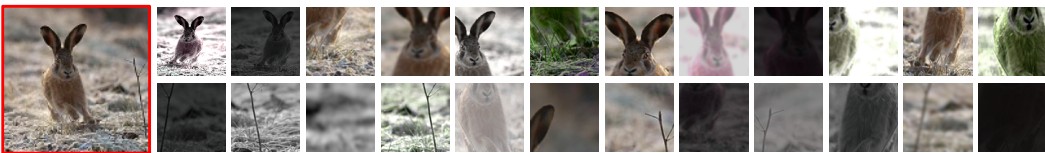

Figure 1: Illustration of OOD samples sampled from distribution $\widetilde{p}$. Figures are ranked according to a descent order of its associated $w_{i,j}$. The biggest image in the red box is the original instance input $x_i$.

## 3 Method

In this section, we reveal that OOD noise is inherently existing in positive view sampling pipelines for a broad range of mainstream SSL approaches. Most importantly, we showcase that these OOD samples are highly structured, and disrupt the downstream tasks with a non-negligible strength. We borrow basic analytic tool from importance sampling to investigate the source of OOD problem specifically for SSL frameworks. We identify an important bias-variance trade-off behind the sampling distribution options for positive views (augmentations), and we analyze the impact of such bias-variance trade-off on the estimator MSE (mean squared error) via rigorous justification. Our model UOTA is motivated to adaptively balance the sampling distribution trade-off via a novel latent variable model, and the proposed method is agnostic to the form of SSL loss. We show that the model is able to effectively suppress the OOD sampling noise for positive samples.

### 3.1 Problem Formulation of SSL

Mainstream SSL approaches, especially contrastive learning methods, often rely on a series transformation functions $x_{i,k} = g(x_i, n_k)$ to produce multiple "views" $x_{i,k}$ of input instance $x_i$. The $\{n_k : k = 1, \ldots, K\}$ is the support of an "augmentation nuisances variable" $n$, representing a group of transformations such as flipping, scaling, cropping, or other augmentations acting on $x_i$ via function $g$. Subscript $k$ indexes all possible augmentations, $i$ indicates the $i$-th instance (each instance corresponds to a specific image input). Without labeling information, SSL relies on the invariance assumption so that deep feature extractor $z_{i,k} = f(x_{i,k}, \theta)$ is at least insensitive to such group of augmentation nuisances $n_k$ w.r.t. the network parameters $\theta$. Existing work [5, 7, 18, 33] show that, while the deep extractor $f(x_{i,k}, \theta)$ is trained to learn the shared semantics from distorted views $x_{i,k}$, such pretrained network parameters $\theta$ effectively benefit further downstream tasks.

SSL approaches based on such pairwised instance invariance assumptions can be generally summarized as penalizing some form of inconsistency between paired of features $z_{i,j}, z_{i,k}$:

$$\mathcal{L}_{\theta}(x_i, (n_j, n_k)) = L(z_{i,j}, z_{i,k}) \tag{1}$$

where $j, k$ indexes over the augmented views on instance $x_i$. Without loss of generality, we expand the support of random variable $n$ to pairwised $\{(n_j, n_k) : j, k = 1, \ldots, K\}$ to incorporate pairs of augmentation actions. Loss $\mathcal{L}_{\theta}(x_i, (n_j, n_k))$ represents such invariance loss under the pair of augmentation actions $(n_j, n_k)$. Here, the generic form of loss function $L(z_{i,j}, z_{i,k})$ imposes the desired cross-view invariance assumptions between $z_{i,j}$ and $z_{i,k}$ for the $i$-th instance. Take for instance, $L(z_{i,j}, z_{i,k})$ can be defined as contrastive loss in MoCo [18], clustered loss as in SwAV [5], or simply $\ell_2$ norm as in BYOL [15]. Our idealized goal then is to sample from independent latent generative processes $x_i \sim p_x$ and $n \sim p^*$ such that $z_{i,k} = f(x_{i,k}, \theta) = f(g(x_i, n_k), \theta)$ captures the same semantics as from original $x_i$.

Assume we have access to such oracle distribution $p^*$, and with equal sampling probability for $x_i$, the objective function is:

$$\mathcal{L}^* = \mathbb{E}_{x \sim p_x, n \sim p^*} [\mathcal{L}_{\theta}(x, n)] \approx \frac{1}{N} \sum_{i=1}^{N} \sum_{j,k} L(z_{i,j}, z_{i,k}) \, p^*(n_j, n_k). \tag{2}$$

Optimizing Eq. (2) then corresponds to learning an M-estimator [22] with regard to network parameters $\theta$.

### 3.2 Sampling Distribution $p^*$: a Heuristic

Augmented views are the bread and butter for self-supervised learning approaches. Intuitively, the network needs to see diverse augmentations from $p^*(n)$ in order to train for invariance. But in the meanwhile, the learning process needs to also avoid training on those $x_{i,k}$ having excessive

distortions who have drastically deviated from the semantics of the original instance. There exists an implicit trade-off in our mind that controls the strength (hyperparameter) of the augmentation actions. The elusive definition on optimal distribution $p^*(\boldsymbol{n})$ therefore seems to be best considered as a *constrained distribution* offering most data variance *subject to* the least semantic deviation from the original instance.

We illustrate this point in Fig. 1. The left-most picture is the original instance input, whereas the two rows of smaller images display the augmented views generated via the multi-crop augmentation pipeline as in SwAV [5]. Apparently, some of augmented views are highlighting the bunny, whereas the others capture the straw only. Under such generative process of augmentations, there is a good chance that a pair of (bunny, straw) features are imposed to be close in feature space. Such noisy invariance might intuitively disrupt the optimization, leading to suboptimal downstream generalization ability. Actually, the work in [36] has theoretically discussed the generalization ability of contrasive pretraining approaches on downstream tasks: if a negative sample is not from a different class of the positive, the upperbound of error on downstream linear classification is provably guaranteed to increase. Similarly, we can derive from [36] via mild modification that if a pair of constructed positive views are not from the same semantic class, the upperbound of linear classification error in the downstream tasks would also increase.

Nevertheless, in practice, while the optimal desired $p^*(\boldsymbol{n})$ remains unknown and elusive, we usually take it for granted that all generated augmentations are equally good, and we tentatively sample $\boldsymbol{n}$ from some heuristically chosen $\widetilde{p}(\boldsymbol{n})$, a surrogate distribution that needs not coincide with the desired $p^*(\boldsymbol{n})$ having the proper augmentation variance-bias trade-off. And instead of optimizing Eq. (2), we often optimize:

$$\tilde{\mathcal{L}} = \mathbb{E}_{\boldsymbol{x} \sim p_x, \boldsymbol{n} \sim \tilde{p}} \left[ \mathcal{L}_{\boldsymbol{\theta}}(\boldsymbol{x}, \boldsymbol{n}) \right] \approx \frac{1}{N} \sum_{i=1}^{N} \sum_{j,k} L(\boldsymbol{z}_{i,j}, \boldsymbol{z}_{i,k}) \, \tilde{p}(\boldsymbol{n}_j, \boldsymbol{n}_k). \tag{3}$$

As per our discussion above, since we ultimately hope to sample from the desired $p^*(\boldsymbol{n})$ instead of the surrogate distribution $\tilde{p}$, we rewrite Eq. (2) using the basic change of measure rule from the importance sampling technique, and yield:

$$\mathcal{L}^* \approx \frac{1}{N} \sum_{i=1}^{N} \sum_{j,k} L(\boldsymbol{z}_{i,j}, \boldsymbol{z}_{i,k}) \, p^*(\boldsymbol{n}_j, \boldsymbol{n}_k) \approx \frac{1}{N} \sum_{i=1}^{N} \sum_{j,k} L(\boldsymbol{z}_{i,j}, \boldsymbol{z}_{i,k}) \, \tilde{p}(\boldsymbol{n}_j, \boldsymbol{n}_k) \cdot \frac{p^*(\boldsymbol{n}_j, \boldsymbol{n}_k)}{\tilde{p}(\boldsymbol{n}_j, \boldsymbol{n}_k)}. \tag{4}$$

Comparing Eq. (4) and Eq. (3), we notice that if we desire to optimize Eq. (2), whereas we insist sampling from $\widetilde{p}$ instead of $p^*$, we should adjust for this change and compensate each loss term via the multiplicative likelihood ratio $p^*(\boldsymbol{n}_j, \boldsymbol{n}_k)/\tilde{p}(\boldsymbol{n}_j, \boldsymbol{n}_k)$, rather than optimizing Eq. (3). This likelihood ratio detects how much chance we would miss the optimal sampling distribution if instead using the surrogate distribution $\widetilde{p}$. If $p^* = \widetilde{p}$, then the objective function recovers the optimal sampling scheme as in Eq. (2) and no adjustment using likelihood ratio is needed.

In practice, SSL approaches usually freeze the optimization on either feature $\boldsymbol{z}_{i,j}$ or $\boldsymbol{z}_{i,k}$, while only the other one is optimized via backpropagation [15, 18]. Therefore it is also reasonable to simplify the objective $\mathcal{L}_{\boldsymbol{\theta}}(\boldsymbol{x}, (\boldsymbol{n}_j, \boldsymbol{n}_k))$ to $\mathcal{L}_{\boldsymbol{\theta}}(\boldsymbol{x}, \boldsymbol{n}_j)$, and the likelihood ratio $p^*(\boldsymbol{n}_j, \boldsymbol{n}_k)/\tilde{p}(\boldsymbol{n}_j, \boldsymbol{n}_k)$ to $p^*(\boldsymbol{n}_j)/\tilde{p}(\boldsymbol{n}_j)$. In addition, for $\boldsymbol{n} = \text{Id}$ which is the identity transformation, $\mathcal{L}_{\boldsymbol{\theta}}(\boldsymbol{x}_i, \text{Id})$ degenerates to the objective function on the original instance $\boldsymbol{x}_i$ without augmentation, we denote it as $\mathcal{L}_{\boldsymbol{\theta}}(\boldsymbol{x}_i)$.

### 3.3  Sampling Distribution $p^*$: a Lemma

Unfortunately, we do not have access to ratio $p^*/\widetilde{p}$ (as hard as knowing $p^*$). But we could define $w(\boldsymbol{n}) = p^*(\boldsymbol{n})/\widetilde{p}(\boldsymbol{n})$, which is essentially the Radon Nikodym derivative, and we tentatively analyze how $w$ would influence the MSE (mean squared error) of deep estimator. To reach this goal, we extend the result in [6], and have the following proposed Lemma on the MSE of deep estimator:

**Lemma 1** *Define* $\boldsymbol{\theta}_0 = \arg\min_{\boldsymbol{\theta}} \mathbb{E}_x \mathcal{L}_{\boldsymbol{\theta}}(\boldsymbol{x})$, $\boldsymbol{\theta}_G = \arg\min_{\boldsymbol{\theta}} \mathbb{E}_x [\int \mathcal{L}_{\boldsymbol{\theta}}(\boldsymbol{x}, \boldsymbol{n}) dp^*(\boldsymbol{n})]$, *and* $\hat{\boldsymbol{\theta}}_G = \arg\min_{\boldsymbol{\theta}} \mathcal{L}^*$. *Let* $\boldsymbol{V}_0$ *be the Hessian of* $\boldsymbol{\theta} \to \mathbb{E}_x \mathcal{L}_{\boldsymbol{\theta}}(\boldsymbol{x})$ *at* $\boldsymbol{\theta}_0$, *and* $\boldsymbol{V}_G$ *the Hessian of* $\boldsymbol{\theta} \to \mathbb{E}_x [\int \mathcal{L}_{\boldsymbol{\theta}}(\boldsymbol{x}, \boldsymbol{n}) d\widetilde{p}(\boldsymbol{n}) w(\boldsymbol{n})]$ *at* $\boldsymbol{\theta}_G$. *Let* $\boldsymbol{M}_0(\boldsymbol{x}) = \nabla \mathcal{L}_{\boldsymbol{\theta}_0}(\boldsymbol{x}) \nabla \mathcal{L}_{\boldsymbol{\theta}_0}(\boldsymbol{x})^T$, *and* $\boldsymbol{M}_G(\boldsymbol{x}) = \nabla \mathcal{L}_{\boldsymbol{\theta}_G}(\boldsymbol{x}) \nabla L_{\boldsymbol{\theta}_G}(\boldsymbol{x})^T$, *where* $\nabla \mathcal{L}_{\boldsymbol{\theta}_0}(\boldsymbol{x})$ *and* $\nabla \mathcal{L}_{\boldsymbol{\theta}_G}(\boldsymbol{x})$ *are gradients of* $\mathcal{L}_{\boldsymbol{\theta}}(\boldsymbol{x})$ *at* $\boldsymbol{\theta} = \boldsymbol{\theta}_0$ *and* $\boldsymbol{\theta} = \boldsymbol{\theta}_G$ *respectively. Similarly, denote* $\boldsymbol{M}_G(\boldsymbol{x}, \boldsymbol{n}) = \nabla \mathcal{L}_{\boldsymbol{\theta}_G}(\boldsymbol{x}, \boldsymbol{n}) \nabla \mathcal{L}_{\boldsymbol{\theta}_G}(\boldsymbol{x}, \boldsymbol{n})^T$. *Denote the trace*

*of matrix $\boldsymbol{X}$ as $tr(\boldsymbol{X})$. Then, with $C$ a constant invariant of $w$, under mild conditions, we have:*

$$MSE(\hat{\boldsymbol{\theta}}_G) \sim C + \|\boldsymbol{\theta}_G - \boldsymbol{\theta}_0\|_2^2 + \frac{1}{N}\mathbb{E}_x[\int tr(\boldsymbol{V}_G^{-1}(\boldsymbol{M}_G(\boldsymbol{x},\boldsymbol{n}) - \boldsymbol{M}_G(\boldsymbol{x}))\boldsymbol{V}_G^{-1})\, d\widetilde{p}(\boldsymbol{n})w(\boldsymbol{n})] \quad (5)$$

$$+ \frac{1}{N}\mathbb{E}_x\left[tr(\boldsymbol{V}_G^{-1}(\boldsymbol{M}_G(\boldsymbol{x}) - \boldsymbol{M}_0(\boldsymbol{x}))\boldsymbol{V}_G^{-1})\right] \quad (6)$$

$$+ \frac{1}{N}tr((\boldsymbol{V}_G^{-1} - \boldsymbol{V}_0^{-1})Cov_x\nabla\mathcal{L}_{\boldsymbol{\theta}_0}(\boldsymbol{x})(\boldsymbol{V}_G^{-1} - \boldsymbol{V}_0^{-1})) \quad (7)$$

$$- \frac{1}{N}tr(\boldsymbol{V}_G^{-1}\mathbb{E}_x\left[Cov_w\nabla\mathcal{L}_{\boldsymbol{\theta}_G}(\boldsymbol{x},\boldsymbol{n})\right]\boldsymbol{V}_G^{-1}), \quad (8)$$

*where $Cov_w(\nabla\mathcal{L}_{\boldsymbol{\theta}_G}(\boldsymbol{x},\boldsymbol{n}))$ is the covariance matrix of $\mathcal{L}_{\boldsymbol{\theta}}(\boldsymbol{x},\boldsymbol{n})$ at $\boldsymbol{\theta}_G$ under measure $p^*(\boldsymbol{n})$.*

Lemma 1 is very busy. But we can still summarize clear variance-bias trade-off implication from it. The key term in Eq. (8) is the covariance matrix of the gradient of loss w.r.t. augmentation measures. In the special case of one-dimensional $\theta$, this term is reflective of the Fisher information and thus larger values lead to better variance reduction for the estimator $\hat{\boldsymbol{\theta}}_G$ and smaller MSE. This means we desire more diverse data. In the meanwhile, terms in Eq. (5) and (6) measure the difference of squared loss gradient between an augmented view and its original instance. If an augmented view is showing a much different gradient $\boldsymbol{M}_G(\boldsymbol{x},\boldsymbol{n})$ in loss than the original clean instance, i.e., gradient $\boldsymbol{M}_G(\boldsymbol{x})$ (as measured in Eq. (5)); or showing a very different gradient $\boldsymbol{M}_G(\boldsymbol{x})$ than the gradient on original instance $\boldsymbol{M}_0(\boldsymbol{x})$ (as in Eq. (6)), then the biases represented in Eq. (5) combined with (6) will increase, and thus the MSE will also increase.

Lemma 1 also explains our intuition in the previous section: we desire large data variance via sampling from $p^*(\boldsymbol{n})$, such that we can reduce the estimator variance Eq. (8). But data variance is a double-edged sword. A radical sampling distribution on $\boldsymbol{n}$ with excessive distortions can simply contradict the desired prerequisite hypothesis (invariance) and increases estimator bias. Through Lemma 1, we are now clear why we would tune the strength of the augmentations to reach the sweet spot for downstream tasks!

Balancing the bias-variance trade-off to reduce MSE in Lemma 1 is tricky. Fortunately, we still have $w(\boldsymbol{n})$ variable to associate with each term in Eq. (5), Eq. (6) and Eq. (8). Recall, the message from Lemma 1 is: if a certain $\boldsymbol{z}_{i,j}$ under the augmentation action of $\boldsymbol{n}_j$ shows a strongly deviating direction from original instance $\boldsymbol{x}_i$'s semantics, it is potentially not representative of the $i$-th instance at all, it perhaps is signaling an out of distribution outlier, and might increase the estimation bias. The variable $w(\boldsymbol{n})$ should be suppressing the effect of these samples. But what is the semantics that original instance $\boldsymbol{x}_i$ mostly cares, the bunny or the straw? The most efficient way is to let all views vote, and to compute a mean feature $\boldsymbol{\mu}_i$ using these augmented views. This $\boldsymbol{\mu}_i$ feature hopefully represents the original semantics. Eq. (5) also implies that, if there is only a single augmented view available, we can additionally forward the original instance without augmentation (i.e., $\boldsymbol{x}_i$) and use feature $\boldsymbol{z}_i = f(\boldsymbol{x}_i,\boldsymbol{\theta})$ to approximate $\boldsymbol{\mu}_i$ (more discussion in Section 4.2). Note, deep neural networks eventually will fit whatever OOD noises via memorization [1], even if the paired invariance is noisy and wrong. We expect to avoid this happening through the influence of $w$ to at least reduce bias term in Eq. (5) and (6) without sacrificing much data diversity.

### 3.4   Optimizing Ratio $w = p^*/\widetilde{p}$ as a Latent Variable

Under measures $p^*$ and $\tilde{p}$ for $\boldsymbol{n}$, the probabilities of sampling $\boldsymbol{x}_{i,j} = g(\boldsymbol{x}_i,\boldsymbol{n}_j)$, where $g$ is invertible, are $p_x(\boldsymbol{x}_i) \cdot p^*(\boldsymbol{n}_j)$ and $p_x(\boldsymbol{x}_i) \cdot \tilde{p}(\boldsymbol{n}_j)$, respectively, due to independence of sampling procedure of $\boldsymbol{x}_i$ and $\boldsymbol{n}_j$. And the likelihood ratio at $\boldsymbol{x}_{i,j}$ is $(p_x(\boldsymbol{x}_i) \cdot p^*(\boldsymbol{n}_j))/(p_x(\boldsymbol{x}_i) \cdot \tilde{p}(\boldsymbol{n}_j)) = p^*(\boldsymbol{n}_j)/\tilde{p}(\boldsymbol{n}_j) = w(\boldsymbol{n}_j)$. Therefore, to find the desired $w(\boldsymbol{n}_j)$ that reduces the MSE in Lemma 1, it suffices to find likelihood ratio in sampling $\boldsymbol{x}_{i,j}$. By the one-one correspondence between $\boldsymbol{x}_{i,j}$ and $\boldsymbol{z}_{i,j}$, it further boils down to calculating weights $w_{i,j}$ for $\boldsymbol{z}_{i,j}$.

With all of these implications from Lemma 1, we correspondingly define the ratio $w_{i,j}$ as:

$$\frac{p^*}{\widetilde{p}} \propto w_{i,j} = \exp[-(\boldsymbol{z}_{i,j} - \boldsymbol{\mu}_i)^T(\tau\boldsymbol{\Sigma})^{-1}(\boldsymbol{z}_{i,j} - \boldsymbol{\mu}_i)], \quad (9)$$

where the $\boldsymbol{\mu}_i$ and $\boldsymbol{\Sigma}$ are defined as:

$$\boldsymbol{\mu}_i = \frac{1}{M}\sum_j^M \boldsymbol{z}_{i,j}, \qquad \boldsymbol{\Sigma} = \frac{1}{NM}\sum_i^N\sum_j^M(\boldsymbol{z}_{i,j} - \boldsymbol{\mu}_i)(\boldsymbol{z}_{i,j} - \boldsymbol{\mu}_i)^T. \quad (10)$$

Here, $\boldsymbol{z}_{i,j}, j \in \{1, ..., M\}$ are features out of $M$ augmentations on $\boldsymbol{x}_i$. Each $(\boldsymbol{z}_{i,j} - \boldsymbol{\mu}_i)$ measures the difference between a specific augmentation feature $\boldsymbol{z}_{i,j}$ and $\boldsymbol{\mu}_i$. Hyperparameter $\tau$ is the temperature that controls the dependency strength of $w_{i,j}$ on such distance. The distance is then normalized through $\boldsymbol{\Sigma}$ among view augmentations. $\boldsymbol{\Sigma}$ normalizes the distance measurement such that all dimensions in $(\boldsymbol{z}_{i,j} - \boldsymbol{\mu}_i)$ are of similar magnitude, avoiding risking a single dimension dominating the distance measure. Eq. (10) computes the covariance using *all the augmentation of all instances* in the current batch and requires all instances share the same $\boldsymbol{\Sigma}$. This seemingly strong hypothesis is absolutely not a simplification. It is rather intended to unveil an appealing unique signature of self-supervised learning that helps shrink the search region of the parameters: the source of variance on all instances are *only* explained by the same set of augmentation actions. Supervised learning does not enjoy similar assumption, as variance of data comes from wild possibilities across classes. Note the construction of the Radon Nikodym derivative $w_{i,j}$ are also supported by the observations in [1]: real data examples are consistently shown to be easier to fit than noise during the training, i.e., the deep model usually first learns the *simple* and general patterns of the real data before fitting the noise. We wouldn't sacrifice learning true general patterns during the process of suppressing noise.

### 3.5 The Unsupervised OuTlier Arbitration Approach (UOTA)

According to the discussion above, the complete UOTA training procedure of $w_{i,j}$ is described as follows (see pseudocode in the supplementary material). We further normalize the ratio value $w_{i,j}$ across the whole batch of samples into $\bar{w}_{i,j}$:

$$\frac{p^*}{\widetilde{p}} \propto \bar{w}_{i,j} = \frac{w_{i,j}}{\sum_i^N \sum_j^M w_{i,j}}. \tag{11}$$

Our loss function Eq. (4) eventually boils down to:

$$\mathcal{L}_{ours} = \sum_{i=1}^{N} \sum_{j=1}^{M} \bar{w}_{i,j} \mathcal{L}_{\boldsymbol{\theta}}(\boldsymbol{x}_i, \boldsymbol{n}_j). \tag{12}$$

During training, one can replace the placeholder loss $\mathcal{L}_{\boldsymbol{\theta}}(\boldsymbol{x}_i, \boldsymbol{n}_j)$ in Eq. (12) by particular form of losses defined in a variety of SSL approaches such as the contrastive loss in MoCo, pairwise $\ell_2$ loss in BYOL or any alternative losses based on pairwise feature invariance assumption. During each training iteration, we firstly forward the training data $\boldsymbol{x}_i$, and update $\bar{w}_{i,j}$ according to Eq. (9) and Eq. (11), where $\boldsymbol{\mu}_i$ and $\boldsymbol{\Sigma}$ are computed using Eq. (10). The value of $\bar{w}_{i,j}$ is then frozen and we backpropagate the loss defined by Eq. (12) only through feature variable $\boldsymbol{z}_{i,j}$ (depending on the SSL loss), to update the network parameters $\boldsymbol{\theta}$. The whole optimizing procedure is an alternating optimization procedure and iterates between updating the latent variable $\bar{w}_{i,j}$ (Eq. (9) and Eq. (11)) given fixed $\boldsymbol{\theta}$; and updating $\boldsymbol{\theta}$ (backpropagate Eq. (12) given fixed $\bar{w}_{i,j}$). Note, the update of $\bar{w}_{i,j}$ in Eq. (11) is motivated from our Lemma 1 and the importance sampling technique, which conveniently jettisons any threshold selection issues in [25, 26, 50]. Particularly we further have the following theorem:

**Theorem 1** *Given mild condition, the optimization of loss Eq. (12) according to the $w_{i,j}$ definition in Eq. (9) leads to a reduced MSE defined as in Lemma 1 than having constant $\bar{w}_{i,j}$.*

The proof is in the supplementary file. The above theorem implies that UOTA procedure and the associated loss function in Eq. (12) are able to effectively correct the estimator bias. We train a network under UOTA procedure, and present the augmentations in Fig. 1 according to descending order of each associated $w_{i,j}$. Interestingly, most augmentations clearly capturing bunny object has higher $w_{i,j}$ values, whereas the straw is arbitrated as highly possible OOD samples (smaller $w_{i,j}$).

## 4 Empirical Analysis

In this section, we empirically evaluate the effectiveness of the proposed UOTA algorithm and justify correctiveness of Theorem 1. In detail, we replace the term $\mathcal{L}_{\boldsymbol{\theta}}(\boldsymbol{x}_i, \boldsymbol{n}_j)$ in Eq.(12) by specific losses defined by a variety of state-of-the-art SSL methods, i.e., MoCo [18], BYOL [15] and SwAV [5]. We then implement our proposed UOTA algorithm (see supplementary file for applying Eq.(12) to these methods) according to the training procedure in Section 3. For each "X+UOTA" model, we firstly train its baseline X for $N_{warm}$ warm up epochs, and then we resume the "X+UOTA" training till end

Table 1: Performance comparison under different training losses using corresponding architectures. All models pretrained for 200 epochs on ImageNet100 [39]. Test accuracy reported in %. Standard deviation ($\pm x$) is reported with 10 different runs in pretraining the model.

| Model | MoCo-v2 | | MoCo-v2* | | BYOL | | SwAV | |
|---|---|---|---|---|---|---|---|---|
| | Top-1 | Top-5 | Top-1 | Top-5 | Top-1 | Top-5 | Top-1 | Top-5 |
| without UOTA | 73.0 ($\pm 0.1$) | 92.0 ($\pm 0.1$) | 80.3 ($\pm 0.1$) | 95.5 ($\pm 0.1$) | 75.3 ($\pm 0.2$) | 93.4 ($\pm 0.1$) | 81.1 ($\pm 0.2$) | 95.9 ($\pm 0.1$) |
| +UOTA | **74.0** ($\pm 0.2$) | **92.4** ($\pm 0.1$) | **81.4** ($\pm 0.1$) | **95.7** ($\pm 0.1$) | **76.2** ($\pm 0.1$) | **93.7** ($\pm 0.1$) | **82.2** ($\pm 0.2$) | **96.1** ($\pm 0.1$) |

Table 2: Performance comparison under different training losses and using corresponding architectures. All models pretrained for 200 epochs on ImageNet100 dataset.

| Model | Top-1 (%) | Top-5 (%) |
|---|---|---|
| MoCo-v2* | 80.3 | 95.5 |
| +Focal [28] | 79.6 | 95.2 |
| +GCE [50] | 80.8 | 95.5 |
| +MIL-NCE [30] | 80.4 | 95.5 |
| +Debiased [10] | 80.7 | 95.6 |
| +UOTA (Ours) | **81.4** | **95.7** |

Table 3: Performance comparison using different estimate strategies for Covariance and Mean in Eq. (10) of UOTA. All models pretrained for 200 epochs on ImageNet100 dataset.

| Model | Covariance | Mean | Top-1 (%) | Top-5 (%) |
|---|---|---|---|---|
| MoCo-v2* | / | / | 80.3 ($\pm 0.1$) | 95.5 ($\pm 0.1$) |
| +UOTA | $\Sigma = I$ | $\mu_i$ | 80.7 ($\pm 0.2$) | 95.5 ($\pm 0.1$) |
| +UOTA | Local $\Sigma_i$ | $\mu_i$ | 80.7 ($\pm 0.2$) | 95.6 ($\pm 0.1$) |
| +UOTA | Global $\Sigma$ | $z_i$ | 81.3 ($\pm 0.1$) | 95.6 ($\pm 0.1$) |
| +UOTA | Global $\Sigma$ | $\mu_i$ | **81.4** ($\pm 0.1$) | **95.7** ($\pm 0.1$) |

(see supplementary file for $N_{warm}$ values used for each experiment). Total number of training epochs of X+UOTA including warm up is the same as that for training X. We follow the standard evaluation protocols as in [18], and evaluate the algorithms as follows: (1) Linear evaluation on frozen pretrained features via UOTA or other various methods on both ImageNet100 [39] and ImageNet1K [11]; (2) Extensive ablation studies against hyperparmeter variations; (3) More downstream task accuracy (i.e., object detection, instance segmentation and keypoint detection) on MS COCO dataset [29] by finetuning pretrained network obtained via UOTA or other SSL approaches. Without specification, all tables with standard deviation ($\pm x$) are reported with 10 different runs pretraining the model.

## 4.1 Evaluating the Unsupervised Features on ImageNet100

In this section, we compare the result of various SSL approaches pretrained with the ResNet-18 Network [20] on ImageNet100 dataset, a subset of ImageNet1K dataset. This is because the suitable data scale allows us to reimplement and frequently tune all the models to reach their best performances. On ImageNet100, we pretrain all the approaches for 200 epochs with a batch size 128. We use each approach's default optimizer (e.g., LARS [45], SGD) and architecture according to their official codes. Following the protocol in [18], we firstly pretrain the ResNet-18 with various SSL losses. We then train linear classifier on frozen features out of each pretrained ResNet-18 to evaluate the classification accuracy on test data. For detailed implementation of all approaches in this section, please see supplementary material.

**Implementing UOTA on various SSL algorithms.** In this section, we implement a variety of SSL algorithms and plug the corresponding losses into Eq. (12) to pretrain the network. We respectively evaluate the linear classification performances on the original baseline models in row "without UOTA" of Table 1, and we present each baseline's performance with UOTA approach in the row "+UOTA". The model MoCo-v2* is an additional baseline we implemented by applying the multi-crop augmentation (8 views $2 \times 224 + 6 \times 96$) and other training techniques from SwAV to MoCo-v2 (LARS optimization, threshold, $lr$ and etc., see supplementary file). Therefore, Table 1 actually covers a broad type of SSL approaches: plain contrasive loss based approach (MoCo-v2); non-contrastive loss with cross view $\ell_2$ norm penalty (BYOL); cluster based multi-view contrastive loss (SwAV, 8 views); and finally contrastive loss with additional augmented views that are trained simultaneously (MoCo-v2*). As Table 1 demonstrates, UOTA is showing evident OOD removal effect with robust performance boost (always $\geq 0.9\%$ gain) against the potential OOD samples in the training data.

**Comparison to other related works.** We compare with other existing OOD detection techniques, some of which originally proposed for supervised learning scenarios. Focal loss [28] is a typical algorithm overweighting hard samples during training; In contrast, GCE [50] is shown to be robust against label noises for the supervised learning similar to self-paced raining procedure [25]. MIL-NCE [30] loss is particularly designed for caption-frames misalignments issues in narrated videos, and is tailored for contrastive loss. Debiased contrastive learning [10] (abbreviated in as Debiased in Table 2) only considers removing the sampling bias from the negative samples. In this section, we

Table 4: All models pretrained for 200 epochs on ImageNet100 dataset. MoCo-v2 and MoCo+ UOTA are trained according to setup in [37]. NDA result is directly from [37].

| Model | Top-1 (%) | Top-5 (%) |
|---|---|---|
| MoCo-v2 | 69.7 | 90.0 |
| MoCo-v2+NDA | 70.0 | / |
| MoCo-v2+UOTA | 70.6 | 90.2 |

Table 5: The 2 class (OOD vs Non-OOD) linear classification accuracy on ImageNet100 data. Group numbers index different strength of augmentations detected by $\bar{w}_{ij}$.

| Model/Group | 1 | 2 | 3 | 4 | 5 | 6 |
|---|---|---|---|---|---|---|
| Training acc. | 99.9 | 97.0 | 92.2 | 87.9 | 86.5 | 51.5 |
| Test acc. | 99.8 | 96.3 | 91.1 | 86.7 | 84.9 | 49.7 |

use MoCo-v2* (defined in previous paragraph) as the baseline model, and implement all the relevant approaches on MoCo-v2* if applicable. We defer the implementation details into the supplementary material. In Table 2, MoCo-v2*+UOTA has achieved the best test accuracy across all the algorithms. Focal loss overweights samples that has the highest loss, thus fails when compared with the baseline model. In contrast, MIL-NCE does not adaptively update the sampling distribution depending on the data, therefore showing an inferior performance than ours. This verifies that OOD issue is indeed a unique issue for the SSL approaches having unique embedded noise structures compared to conventional supervised learning approaches. UOTA outperforms "Debiased", which at least shows that addressing the noise issue in positive samples is equally important as in negative samples.

While UOTA tries to mute the intrinsic OOD samples from positive, the NDA approach in [37] even considers manually generating OOD samples as negatives. These two methods differ in motivation, but both would be provably shown to result in reduced error bound on the downstream linear classification tasks, as per the discussion in paper [36]. We compare scores reported in [37] by following the ResNet50 training setups of Table 6 in [37]. The results are reported in Table 4, showing the effectiveness of both methods while UOTA is relatively better.

## 4.2 Ablation Studies

In this section, we examine the impact of different hyperparamters in the UOTA framework on ImageNet100. We evaluate different settings on linear classification task as defined in Section 4.1, with the same pretraining set up for UOTA as described as in Section 4.1. The impact of $\tau$, $N_{warm}$ and other hyperparameters of UOTA is included in supplementary material.

**Impact of crop-min strength in augmentation** We first investigate the effect of the crop-min value. The hyperparameter crop-min defines the minimal size of an augmentation via random cropping from the original instance. According to Lemma 1, there should be a sweet point of crop-min value best balancing the trade-off for the estimator MSE. Interestingly, we spot the expected sweetpoint for the MoCo-v2 baseline at value of 0.16, as in Fig. 2(a). Notice that MoCo-v2+UOTA has an evident effect in flattening the influence of crop-min against the accuracy. This well corroborates the efficient role of UOTA, which adaptively adjust for the sampling distribution to combat the OOD samples. In this way, UOTA guards the SSL approaches without suffering too much OOD noise.

**Impact of number of views for each instance**. We adopt the MoCo-v2* framework as the baseline to associate with UOTA. Fig. 2(b) shows, as the view numbers for MoCo-v2* increase, the UOTA approach consistently outperforms MoCo-v2* by a clear margin. This advantage is even slightly promoted as view number increases. We speculate that as the diversity of views grows, both algorithm benefit from increased data variance and reduced estimator variance through Lemma 1, while UOTA enjoys better performance gain via Theorem 1.

**Impact of covariance estimate**. In Section 3.4, we impose the constraint that all instances share the identical covariance. This implicit constraint is critical to practically reduce MSE defined in Lemma 1. In this section, we justify our hypothesis by comparing with two baselines on MoCo-v2* (8 views). The "local $\Sigma_i$" baseline computes a unique $\Sigma_i$ for each instance $i$ by only involving its own augmentations. "$\Sigma = I$" baseline further removes the whitening step (effectively a squared euclidean distance) when evaluating the feature distance (between $z_{i,j}$ and $\mu_i$). In contrast, "global $\Sigma$" is our default option for UOTA as defined in Eq. (10), i.e., "global $\Sigma$" is estimated via all augmentations of all instances (all instances share the same $\Sigma$). As Table 3 shows, "global $\Sigma$" outperforms both "local $\Sigma$" and "$\Sigma = I$", which demonstrates the benefit of proposed constraint.

**Impact of mean estimate**. In Section 3.3, we hypothesize that $\mu_i$ is around the value of original $z_i$, as original $z_i$ feature can be regarded as effectively integrating out the augmentation nuisance

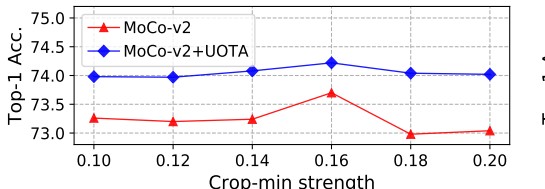 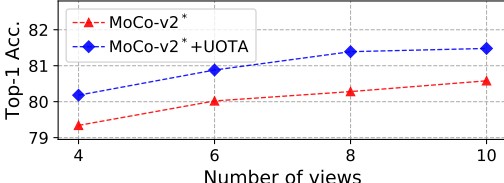

(a) Impact of crop-min strength in augmentation.     (b) Impact of number of views for each instance.

Figure 2: Impact of several hyperparameters of UOTA: (a) crop-min strength, (b) number of views.

variable $n$ given certain $z_i$. We therefore examine the UOTA performance under the MoCo-v2* (8 views) framework using these two different mean estimates. "$z_i$" model means we replace the $\mu_i$ by using the original feature (without augmentation) $z_i$ in Eq. (10). The cost of this replacement is an extra forward of original $x_i$, though. As observed in Table 3, there is slight difference between the two models, showing effectiveness using average $\mu_i$ to represent $z_i$.

### 4.3 Evaluating the Unsupervised Features on ImageNet1K

**Implementation details.** In this section, we compare the result of various SSL approaches pretrained on the ResNet-50 [20]. Following the protocol in [18], we firstly pretrain the ResNet-50 with various SSL losses. We then implement further supervised downstream tasks by either finetuning on the pretrained network, or by directly using the frozen feature out of the pretrained network depending on the task. In this section, we implement the proposed UOTA algorithm by using the SwAV loss (multi-crop with 8 views $2 \times 224 + 6 \times 96$), the exactly architecture and augmentation pool as in [5] to optimize Eq. (12). We train all relevant algorithms on 4 V100 GPUs, with a batch size 256. For other training details, please refer to supplementary task.

**Evaluating the linear classification task on frozen features.** We further train linear classifier on frozen features out of each pretrained ResNet-50 network to evaluate the classification accuracy on test data. As Table 6 displays, our proposed UOTA algorithm (associated with the SwAV approach) has achieved the best Top-1 accuracy (73.5%) and Top-5 accuracy (91.8%) across all the presented SSL pretraining algorithms. Particularly, the proposed SwAV+UOTA pretrained for 200 epochs even surpasses SimCLR [7] and Barlow Twin [47] trained for 1000 epochs. Besides, the SwAV+UOTA framework outperforms the original SwAV baseline by 0.8% accuracy. In the meanwhile, the total training time for SwAV+UOTA is **181.5** hours, with only negligible 2 hours more than SwAV's training time **179.2** hours on 4 V100 GPUs. These results demonstrate clear evidence of our proposed hypothesis: OOD samples broadly exist in the training data. Importantly, the OOD issue is not vanishing as the the number of instances in training significantly grows (much more than in ImageNet100). Fortunately, our lightweight UOTA procedure is able to modify the sampling distribution and reduce the estimator bias, guaranteed by our Lemma 1 and Theorem 1. We think this 0.8% improvement effectively raises a warning flag to the community, and it verifies that OOD samples has a clear and non-negligible impact that should be attended to.

**Binary classification between OOD data and clean data.** In this section, we empirically verify that OOD samples and original image are linear separable on frozen features learned by UOTA. For this purpose, we choose ImageNet100 training data as training set and ImageNet1K val set (50,000 images) as test set. Each of the training data is randomly augmented into 5 views. We reorder all the samples according to their computed $\bar{w}_{ij}$ value in an ascending order (Group 1 contains the lowest $\bar{w}_{ij}$ valued samples, showing strongest sign of OOD sample; Group 5 has the highest $\bar{w}_{ij}$ samples, best resembling the semantic of the original training sample). We sequentially split such reordered augmentation set into 5 groups with equal sample size. Different groups then reflect the detected semantic uncertainty via UOTA. The test data are augmented in the same way and split into 5 groups, too. We then train the binary classifier on top of the frozen pre-trained model (SwAV+UOTA model producing Table 6), by respectively treating each Group $n$ as class 1 (OOD class), with clean data considered as class 0 (class 0 and 1 have equal number of training data). Group 6 is a baseline with equal size of clean data in two subsets representing class 1 and class 0 respectively. More training details are included in the supplementary file. As Table 5 shows, the binary linear classifier between OOD and clean data on their frozen features always show clear statistical significance and better separability against the reference baseline Group 6. Notably, as the data exhibits weaker OOD signs (e.g., Group 5), the training becomes harder and harder, returning lower training accuracy, while the OOD samples still demonstrate much stronger linear separability against the baseline model Group 6.

Table 6: Accuracy of linear classification model on ImageNet1K. Bold numbers are the best performance among models trained for 200 epochs. Numbers **(+x%)** denotes additional gain compared to the baseline model (i.e., SwAV here in the table) without UOTA approach. [†] denotes results represented from [3, 9]. [‡] means results of our reproduced reproduced based on SwAV official code.

| Method | Epochs | Batch size | Top-1 (%) | Top-5 (%) |
|---|---|---|---|---|
| CPC v2 [21] | 200 | 512 | 63.8 | 85.3 |
| CMC [39] | 240 | / | 64.8 | 86.1 |
| MoCo [18] | 200 | 256 | $60.6^{\dagger}$ | $83.1^{\dagger}$ |
| MoCo v2 [8] | 200 | 256 | $67.6^{\dagger}$ | $88.0^{\dagger}$ |
| JCL [3] | 200 | 256 | 68.7 | 89.0 |
| SimCLR [7] | 1000 | 4096 | 69.3 | 89.0 |
| SimSiam [9] | 200 | 256 | 70.0 | / |
| InfoMin Aug. [40] | 200 | 256 | 70.1 | 89.4 |
| BYOL [15] | 200 | 4096 | $70.6^{\dagger}$ | / |
| Barlow Twin [47] | 1000 | 2048 | 73.2 | 91.0 |
| SwAV [5] | 200 | 256 | $72.7^{\ddagger}$ | $91.5^{\ddagger}$ |
| SwAV+UOTA (Ours) | 200 | 256 | **73.5 (+0.8%)** | **91.8 (+0.3%)** |

Table 7: Performance on downstream tasks: object detection [35] (left), instance segmentation [19] (middle) and keypoint detection [19] (right). Accuracy in %. All models pretrained 200 epochs and finetuned on MS COCO with $1 \times$ schedule.

| Model | Faster R-CNN + R50-FPN | | | Mask R-CNN + R50-FPN | | | Keypoint R-CNN + R50-FPN | | |
|---|---|---|---|---|---|---|---|---|---|
| | $AP^{bb}$ | $AP^{bb}_{50}$ | $AP^{bb}_{75}$ | $AP^{mk}$ | $AP^{mk}_{50}$ | $AP^{mk}_{75}$ | $AP^{kp}$ | $AP^{kp}_{50}$ | $AP^{kp}_{75}$ |
| random | 30.1 | 48.6 | 31.9 | 28.5 | 46.8 | 30.4 | 63.5 | 85.3 | 69.3 |
| supervised | 38.2 | 59.1 | 41.5 | 35.4 | 56.5 | 38.1 | 65.4 | 87.0 | 71.0 |
| MoCo-v1[18] | 37.1 | 57.4 | 40.2 | 35.1 | 55.9 | 37.7 | 65.6 | 87.1 | 71.3 |
| MoCo-v2 [8] | 37.6 | 57.9 | 40.8 | 35.3 | 55.9 | 37.9 | 66.0 | 87.2 | 71.4 |
| JCL [3] | 38.1 | 58.3 | 41.3 | 35.6 | 56.2 | 38.3 | 66.2 | 87.2 | **72.3** |
| InfoMin Aug. [40] | / | / | / | **36.7** | 57.7 | **39.4** | / | / | / |
| SwAV | 38.5 | 60.5 | 41.4 | 36.3 | 57.7 | 38.9 | 65.6 | 86.9 | 71.6 |
| SwAV+UOTA | **39.0** | **61.0** | **42.0** | **36.7** | **58.4** | **39.4** | **66.3** | **87.4** | **72.3** |

**Finetuning on pretrained network for more downstream tasks.** In this section, we evaluate the presented SSL algorithms by finetuning the pretrained ResNet50 equipped with FPN [27] on more downstream tasks. Table 7 respectively reports the performance on object detection, instance segmentation and keypoint detection tasks. These tasks, to some extent, evaluate the generalization capability of the pretrained features when transferred to other tasks and training datasets. According to Table 7, SwAV+UOTA still outperforms its baseline model SwAV on all evaluation metrics for the object detection and instance segmentation tasks. Notably, SwAV baseline drops drastically on the keypoint detection tasks in comparison to MoCo-v2. We speculate that the multi-resolution multi-crop augmentation has introduced more OOD samples into SwAV than in MoCO-v2, leading to vague semantic feature that is critical for successful key-point detection tasks. This well reveals that, even if we further finetune the pretrained network using alternative dataset, SwAV+UOTA is still able to benefit tasks by suppressing the OOD samples and balancing the estimator bias-variance trade-off during the pretrain. The advantage of UOTA is not erased due to the finetuning. Table 7 concludes that the down-weighted OOD samples through training using UOTA approach successfully forced the network to focus more on the true invariant semantics shared across multi-views, which plays pivotal importance on downstream task performance.

## 5   Conclusion

In this paper, we explore the potential OOD noise issue for SSL approaches. Most importantly, we demonstrate that these OOD samples are highly structured, and are interfering the downstream tasks with a non-negligible impact. We use importance sampling technique to investigate the source of OOD problem via estimator MSE decomposition, and we identify an important bias-variance trade-off behind the sampling distribution options for positive views. We propose a useful latent variable model UOTA to effectively balance such trade-off. Empirical study well corroborates our hypothesis and shows strong advantage of our UOTA approach over the state-of-the-art SSL approaches.

**Social Impact and Limitation.** With our SSL approach, one can conveniently construct her own pretraining dataset by random crawling data. This is economic and beneficial for research purpose, but would potentially risk privacy and license issues. Also, given the easy access to data via unsupervised learning, one might risk abusing the use of AI technique and applying in appropriate occasions. Our approach also requires tuning extra hyperparameters depending on dataset.

## Acknowledgments and Disclosure of Funding

Funding in direct support of this work: the National Key R&D Program of China under Grant No. 2020AAA0108600.

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
