# Improving Self-supervised Learning with Automated Unsupervised Outlier Arbitration Supplementary File

**Yu Wang**[1]  **Jingyang Lin**[2]  **Jingjing Zou**[3]  **Yingwei Pan**[1]  **Ting Yao**[1]  **Tao Mei**[1]

[1] JD AI Research, Beijing, China
[2] Sun Yat-sen University, Guangzhou, China
[3] University of California, San Diego, USA

{feather1014, yung.linjy}@gmail.com, j2zou@ucsd.edu
{panyw.ustc, tingyao.ustc}@gmail.com, tmei@jd.com

## 1 Introduction

The file is a supplementary file of paper [18]. This file is organized as follows: Section 2 presents the pseudo-code of UOTA approach. Section 3 discusses the role of $\tau$ in UOTA. Section 4 shows the proof of Lemma 1 and Theorem 1 proposed in the main file. Section 5, Section 6 and Section 8 explain more implementation details of the empirical implementation. We use "M" or "S" to distinguish contents in the main file or in the supplementary file. For instance, we refer to specific figure/table/equation/section in the main file as Fig. M.x/Table M.x/Section M.x/Eq. (M.x)/. We use Fig. S.x/Table S.x/Section S.x/Eq. (S.x) to indicate contents in supplementary file. Our code is available at: https://github.com/ssl-codelab/uota.

## 2 Pseudo-code of UOTA Approach in PyTorch style

```
# f: encoder f
# g: encoder g
# L: various self-supervised objectives
# tau: temperature of UOTA
# M: number of sub-losses for each instance pair

for x in loader: # load a mini-batch x with n samples
    multi_x0, multi_x1 = aug(x), aug(x)   # random augmentations
    multi_z = f(multi_x0) # trainable features z_ij: (M*N,D)
    multi_k = g(multi_x1) # features k_ij: (M*N,D)

    with no_grad(): # compute w based on trainable features z_ij
        w = W(multi_z.reshape(M, N, D), tau)

    loss = mean(w * L(multi_z, multi_k)) # Eq.(M.12)
    loss.backward()
    update(f, g)

# UOTA: compute w based on trainable features z_ij
def W(multi_z, tau):
  M, N, D = multi_z.shape
  z_mu = mean(multi_z, dim=0, keepdim=True)     # Eq.(M.10): (1,N,D)
  z_delta = (multi_z - z_mu).reshape(-1, D).T   # Eq.(M.10): (D,M*N)
  sigma = mm(z_delta, z_delta.T) / N            # Eq.(M.10): (D,D)

  tmp = mm(z_delta.T, inverse(sigma) / tau)
  w = exp(-mm(tmp, z_delta)).diagonal(-2, -1)   # Eq.(M.9) : (M*N,)
  w = w / w.sum() * (M * N)                     # Eq.(M.11): (M*N,)
  return w
```

35th Conference on Neural Information Processing Systems (NeurIPS 2021).

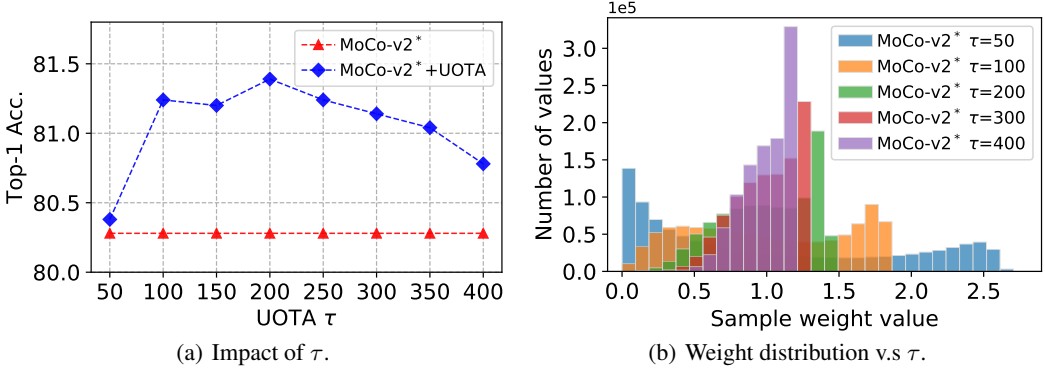

(a) Impact of $\tau$.  (b) Weight distribution v.s $\tau$.

Figure 1: Illustrations of hyperparameter $\tau$: (a) Impact of $\tau$, (b) Weight distribution v.s $\tau$.

## 3 Ablation Study on Hyperparamter $\tau$

Owing to page limit of the main paper, we present the supplementary discussion on hyperparamter $\tau$ here. We follow the same experimental setup as in Section M.4.1, and pretrain a ResNet18 on the ImageNet100 dataset. We run UOTA on top of the baseline model MoCo-v2* with 8 views, and evaluate UOTA's performance on linear classification task. Please see Section S.5 for more training details and experimental setups on this task. All the scores are reported in Top-1 accuracy.

Before seeing any curves, we shall understand that, the $\tau$ in our Eq. (M.9) and Eq. (M.11) play a similar role as the temperature defined in [8, 10, 16, 19], i.e., $\tau$ controls the concentration of the probability $\bar{w}_{i,j}$. A relatively large $\tau$ will dilute the effect of distance metric $(z_{i,j} - \mu_i)^T \Sigma^{-1}(z_{i,j} - \mu_i)$ on $\bar{w}_{i,j}$, making the distribution of $\bar{w}_{i,j}$ more even and flat across samples. A relatively small $\tau$ will force the $\bar{w}_{i,j}$ to be more sensitive to the changes in distance $(z_{i,j} - \mu_i)^T \Sigma^{-1}(z_{i,j} - \mu_i)$, making the distribution of $\bar{w}_{i,j}$ asymptotically to "one hot" (probability concentrating around the smallest $(z_{i,j} - \mu_i)^T \Sigma^{-1}(z_{i,j} - \mu_i)$ value). In the meanwhile, the value of $\tau$ also correspondingly determines the "effective number of samples" [14] active in training. As the value of $\tau$ drops below some certain point, we start to lose effective number of samples, according to [14]. Reduced "effective number of samples" potentially will hamper the performance when compared to plain MoCo-v2* formulation, as we are losing views. However, if $\tau$ becomes too big, the $\bar{w}_{i,j}$ tends to become even across samples (concentrating around constant 1), making the outlier arbitration capability of UOTA vanishing.

In Fig. S.1(a), we firstly illustrate the performance curve of UOTA against the variation of $\tau$. As can be seen, the suitable range of $\tau$ is reasonably wide. As $\tau$ varies between 50 and 400, UOTA+MoCo-v2* surpasses the plain MoCo-v2* by a clear margin. But as $\tau$ further increases beyond 350, performance gradually approaches the similar level as plain MoCo-v2*, due to the fact that $w_{i,j}$ across all the samples $x_{i,j}$ are approaching a constant value, which makes all samples equally important. The performance of UOTA+MoCo-v2* therefore effectively approaches that of a plain MoCo-v2* framework. Conversely, when $\tau$ decreases to be smaller than 100, the performance also drastically drops, due to the fact that the effective number in the training is potentially less than MoCo-v2*'s 8 views.

In Fig. S.1(b), we illustrate the distribution of $\bar{w}_{i,j}$ (according to Eq. (M.11)) across training samples under the variation of $\tau$. For illustration purpose, we scale up the value of each $\bar{w}_{i,j}$ by multiplying MN, which removes the normalization scales, and helps better qualitative analysis on distribution of $w_{i,j}$ in an interpretable way, i.e., we define $\hat{w}_{i,j} = MN\bar{w}_{i,j}$ and plot value $\hat{w}_{i,j}$ in Fig. S.1(b). Specifically, we compute the $\bar{w}_{i,j}$ values using Eq. (M.11) at the 8 views plain MoCo-v2* model's $50^{th}$ training epoch, and then plot the histograms of $\hat{w}_{i,j}$ on 126,689 number of training samples of ImageNet100. Fig. S.1(b) well displays the concentration behavior of $\hat{w}_{i,j}$ via tuning the value of $\tau$: as $\tau$ increases (e.g., 400), all the $\hat{w}_{i,j}$ values tend to concentrate around the constant value of 1 (all samples are equally important), showing vanishing effect of the distance metric $(z_{i,j} - \mu_i)^T \Sigma^{-1}(z_{i,j} - \mu_i)$. On the contrary, as $\tau$ gradually decreases, $\hat{w}_{i,j}$ values spread out and show better diversity and variance reflective of the sample-wise $(z_{i,j} - \mu_i)^T \Sigma^{-1}(z_{i,j} - \mu_i)$

values. But when $\tau$ further decreases below 50, $w_{i,j}$ becomes more and more one-hot like and assigns more probability mass to 0 values (losing more effective sample sizes). This also echos the observation in Fig. S.1(a).

# 4 Proof

For the proof of Lemma 1 and Theorem 1, we are assuming the same regularity conditions including Assumptions A and B on the loss function $\mathcal{L}$ as in [3]. Note without ambiguity $w_{i,j}$ in this section denotes the properly normalized weights.

## 4.1 Proof of Lemma 1

Given that $\boldsymbol{\theta}_G = \arg\min_\theta \mathbb{E}_x[\int \mathcal{L}_{\boldsymbol{\theta}}(\boldsymbol{x}, \boldsymbol{n}) dp^*(\boldsymbol{n})]$ and

$$\hat{\boldsymbol{\theta}}_G = \arg\min_\theta \mathcal{L}^* = \arg\min_\theta \frac{1}{N} \sum_{i=1}^N \sum_{k=1}^M \mathcal{L}_{\boldsymbol{\theta}}(\boldsymbol{x}_i, \boldsymbol{n}_k) w_{i,k},$$

with Theorem 5.23 of [17], we have

$$\sqrt{N}(\hat{\boldsymbol{\theta}}_G - \boldsymbol{\theta}_G) = -\boldsymbol{V}_G^{-1} \cdot \frac{1}{\sqrt{N}} \sum_{i=1}^N \sum_{k=1}^M \nabla \mathcal{L}_{\boldsymbol{\theta}_G}(\boldsymbol{x}_i, \boldsymbol{n}_k) w_{i,k} + o_p(1),$$

where $o_p(1) \to 0$ in probability as $N \to \infty$. Therefore

$$\text{MSE}(\hat{\boldsymbol{\theta}}_G) = \|\boldsymbol{\theta}_G - \boldsymbol{\theta}_0\|^2 + \frac{1}{N} \text{tr}(\boldsymbol{V}_G^{-1} \cdot \text{Cov}_x \left[\nabla \int \mathcal{L}_{\boldsymbol{\theta}_G}(\boldsymbol{x}, \boldsymbol{n}) w(\boldsymbol{n}) d\tilde{p}(\boldsymbol{n})\right] \cdot \boldsymbol{V}_G^{-1}), \quad (1)$$

where $\nabla$ denotes the gradient with respect to $\boldsymbol{\theta}$ and $\boldsymbol{V}_G$ denotes the Hessian matrix $\text{Hess}(\mathbb{E}_x[\int \mathcal{L}_{\boldsymbol{\theta}}(\boldsymbol{x}, \boldsymbol{n}) w(\boldsymbol{n}) d\tilde{p}(\boldsymbol{n})])$ at $\boldsymbol{\theta}_G$. Note the term $\text{Cov}_x[\nabla \int \mathcal{L}_{\boldsymbol{\theta}_G}(\boldsymbol{x}, \boldsymbol{n}) w(\boldsymbol{n}) d\tilde{p}(\boldsymbol{n})]$ can be rewritten as in the proof of Lemma 1 of [3]:

$$\text{Cov}_x \left[\nabla \int \mathcal{L}_{\boldsymbol{\theta}_G}(\boldsymbol{x}, \boldsymbol{n}) w(\boldsymbol{n}) d\tilde{p}(\boldsymbol{n})\right] - \mathbb{E}_x \mathbb{E}_{p*} \boldsymbol{M}_G(\boldsymbol{x}, \boldsymbol{n})$$

$$= \mathbb{E}_x(\nabla \mathbb{E}_p^*(\mathcal{L}_{\boldsymbol{\theta}_G}(\boldsymbol{x}, \boldsymbol{n})) \cdot \nabla \mathbb{E}_p^*(\mathcal{L}_{\boldsymbol{\theta}_G}(\boldsymbol{x}, \boldsymbol{n}))^T) - \mathbb{E}_x(\nabla \mathbb{E}_p^*(\mathcal{L}_{\boldsymbol{\theta}_G}(\boldsymbol{x}, \boldsymbol{n}))) \cdot \mathbb{E}_x(\nabla \mathbb{E}_p^*(\mathcal{L}_{\boldsymbol{\theta}_G}(\boldsymbol{x}, \boldsymbol{n})))^T$$

$$- \mathbb{E}_x(\mathbb{E}_{p^*}(\nabla \mathcal{L}_{\boldsymbol{\theta}_G}(\boldsymbol{x}, \boldsymbol{n}) \cdot \nabla \mathcal{L}_{\boldsymbol{\theta}_G}(\boldsymbol{x}, \boldsymbol{n})^T))$$

$$= -\mathbb{E}_x(\text{Cov}_w \nabla \mathcal{L}_{\boldsymbol{\theta}_G}(\boldsymbol{x}, \boldsymbol{n})).$$

The last equation holds since $\mathbb{E}_x(\nabla \mathbb{E}_{p^*}(\mathcal{L}_{\boldsymbol{\theta}_G}(\boldsymbol{x}, \boldsymbol{n}))) = \boldsymbol{0}$ by changing the order of $\nabla$ and $\mathbb{E}_x$ under regularity conditions on $\mathcal{L}$. The rest of the results can be proved by decomposing $\mathbb{E}_x \mathbb{E}_{p*} \boldsymbol{M}_G(\boldsymbol{x}, \boldsymbol{n})$ further in the manner of proof of Lemma 1 in [3].

## 4.2 Proof of Theorem 1

By combining Eq. (M.5) - (M.8) in Lemma 1, the mean squared error of $\hat{\boldsymbol{\theta}}_G$ can be further simplified as

$$\text{MSE}(\hat{\boldsymbol{\theta}}_G) \sim C + \|\boldsymbol{\theta}_G - \boldsymbol{\theta}_0\|^2 + \frac{1}{N} \text{tr}(\boldsymbol{V}_G^{-1} \cdot \text{Cov}_x \left[\nabla \mathbb{E}_{p^*}(\mathcal{L}_{\boldsymbol{\theta}_G}(\boldsymbol{x}, \boldsymbol{n}))\right] \cdot \boldsymbol{V}_G^{-1}), \quad (2)$$

where $C$ is a constant invariant of the choice of $\{w_{i,j}\}$. Without loss of generality, assume the model parameters $\boldsymbol{\theta}$ are orthogonalized so that Eq. (S.2) can be further simplified as only depending on

$$\|\boldsymbol{\theta}_G - \boldsymbol{\theta}_0\|^2 + \frac{1}{N} \frac{1}{\boldsymbol{V}_G^2} \mathbb{E}_x \left[(E_{p^*}(\mathcal{L}'_{\boldsymbol{\theta}_G}(\boldsymbol{x}, \boldsymbol{n})))^2\right],$$

where $\mathcal{L}'_{\boldsymbol{\theta}_G}(\boldsymbol{x}, \boldsymbol{n})$ denotes the diagonal gradient matrix of $\mathcal{L}$ with respect to $\boldsymbol{\theta}$ and $\boldsymbol{V}_G^2$ is the diagonal Hessian matrix at $\boldsymbol{\theta}_G$.

Assume the loss function $\mathcal{L}$ satisfies a symmetric condition: there exists an interval of $(\boldsymbol{\theta}_1, \boldsymbol{\theta}_2)$ such that $\boldsymbol{\theta}_0, \boldsymbol{\theta}_G \in (\boldsymbol{\theta}_1, \boldsymbol{\theta}_2)$. Suppose for all $\boldsymbol{\theta} \in (\boldsymbol{\theta}_1, \boldsymbol{\theta}_2)$, $\mathcal{L}_{\boldsymbol{\theta}}(\boldsymbol{x}_i, \boldsymbol{n}_j) = \mathcal{L}_{\boldsymbol{\theta}}(\boldsymbol{x}_i, \boldsymbol{n}_k)$ whenever

the squared Mahalanobis distance of the corresponding features $z_{i,j}$ and $z_{i,k}$ are equal: $(z_{i,j} - \mu_i)^T \Sigma^{-1}(z_{i,j} - \mu_i) = (z_{i,k} - \mu_i)^T \Sigma^{-1}(z_{i,k} - \mu_i)$. As a result, for $\theta_G \in (\theta_1, \theta_2)$, the second term in Eq. (S.2) can be written as

$$\frac{1}{N} \frac{1}{V_G^2} \mathbb{E}_x \Big[ \Big( \sum_{j=1}^{M} \breve{\mathcal{L}}'_{\theta_G}(d^2(z_{i,j}, \mu_i)) \, w_{i,j} \Big)^2 \Big], \tag{3}$$

where $\breve{\mathcal{L}}$ denotes the function as a result of rewriting $\mathcal{L}$ in terms of the squared Mahalanobis distance denoted by $d^2(z_{i,j}, \mu_i)$. Note $\arg\min_\theta \mathbb{E}_x \sum_{j=1}^{M} \breve{\mathcal{L}}_\theta(d^2(z_{i,j}, \mu_i)) w_{i,j} = \arg\min_\theta \mathbb{E}_x[\int \mathcal{L}_\theta(x, n) dp^*(n)] = \theta_G$. Suppose the distribution of the Mahalanobis distance of the generated $\{z_{i,j}\}$ is approximately uniform, and $\mathbb{E}_x\Big[ \Big( \sum_{j=1}^{M} \breve{\mathcal{L}}'_{\theta_G}(d^2(z_{i,j}, \mu_i)) \, w_{i,j} \Big)^2 \Big]$ can be well approximated by $(\int \breve{\mathcal{L}}_{\theta_G}(x)) \breve{w}(x))^2$, where $\breve{w}$ is the representation of $w$ as a function of the Mahalanobis distance. Then for all loss function $\mathcal{L}$ such that $|\mathbb{E}_1(\breve{\mathcal{L}}_{\theta_G}(x))| < |\mathbb{E}_2(\breve{\mathcal{L}}_{\theta_G}(x))|$, where $E_1$ and $E_2$ are expectations under standard exponential and uniform distribution defined on an interval of admissible values of the Mahalanobis distance, the proposed $w_{i,j}$ given in Eq. (M.9) in the main paper decreases Eq. (S.3) comparing to equal weights. Essentially, $z_{i,j}$ with larger Mahalanobis distance is down-weighted to reduce its leverage in the estimating equation (for methods using similar techniques, see, for example, [9]).

Finally, given Assumption B in [3], the squared bias $\|\theta_G - \theta_0\|^2$ is reduced by selecting measure $p(n)$ to approach the invariance $\mathbb{E}_x \int \mathcal{L}_\theta(x, n) dp(n) = \mathbb{E}_x[\mathcal{L}_\theta(x, \text{Id})]$ on interval $(\theta_1, \theta_2)$, as $\|\theta_G - \theta_0\| \sim |\mathbb{E}_x \int \mathcal{L}_{\theta_G}(x, n) dp(n) - \mathbb{E}_x[\mathcal{L}_{\theta_0}(x, \text{Id})]|/l$ where $l$ is square-integrable function not depending on $\theta$. Given our assumption that $w(n)p(n) = p^*(n)$ improves invariance comparing to $\tilde{p}(n)$ with $|\mathbb{E}_x \int \mathcal{L}_\theta(x, n) dp^*(n) - \mathbb{E}_x[\mathcal{L}_\theta(x, \text{Id})]| < |\mathbb{E}_x \int \mathcal{L}_\theta(x, n) d\tilde{p}(n) - \mathbb{E}_x[\mathcal{L}_\theta(x, \text{Id})]|$ for $\theta \in (\theta_1, \theta_2)$, using triangular inequality, we conclude that imposing the weight adjustments with $w$ defined in Eq. (M.9) can effectively reduce the MSE.

## 5 Implementation Details of Section M.4.1

We pretrain a ResNet-18 Network on ImageNet100 by applying Eq. (M.12) to various SSL algorithms. We then evaluate linear classification performance on both original SSL baseline models and UOTA approach on top of these baseline models. For all algorithms presented, we use batch size $N = 128$ and train for 200 epochs. We use the default optimization pipelines used for each specific published method. We are confident that we have tuned the hyperparamters for all the approaches to our best. After we obtain the pretrained model, we then train a supervised linear classifier on top of the frozen representations of ResNet-18, by following protocols implemented in [2, 4, 7, 8].

### 5.1 Optimization for Pretraining SSL Baselines on ImageNet100 (Table M.1)

**UOTA**. We adhere to exactly same optimization pipeline (architecture, optimizer, learning rate, hyperparameters, etc., ) used for training each original baseline model "X" in order to train and optimize each "X+UOTA" model. For each "X+UOTA" model, we firstly train its baseline X for number of $N_{warm}$ warm up epochs, and then we resume the "X+UOTA" training by following the algorithm described as in Section S.2. Total number of training epochs are 200, including warm up epochs. We also tune the hyperparameter $\tau$ defined in Eq. (M.9) for each model. These hyperparameters are illustrated in Table S.1:

Table 1: Hyperparameters for UOTA loss when Producing Table M.1.

| Hyperparamter/Model | MoCo-v2+UOTA | MoCo-v2*+UOTA | BYOL+UOTA | SwAV+UOTA |
|---|---|---|---|---|
| $N_{warm}$ | 20 | 20 | 20 | 30 |
| $\tau$ | 200 | 200 | 250 | 200 |

Here, we would like to further ablate the effect of the resuming point (from which epoch to resume a MoCo-v2 baseline model by adding UOTA, while the total training epochs remains as 200 epochs on ImageNet100 dataset. The reference accuracy of a baseline MoCo-v2 is 73.0%. The ablation is reported in Table S.2.

Table 2: Resuming point ablations. Top 1 accuracy is reported in %.

| $N_{warm}$ | 0 | 10 | 20 | 30 | 50 | 100 | 150 |
|---|---|---|---|---|---|---|---|
| Top 1 | 73.5 | 73.7 | **74.0** | 73.9 | 73.6 | 73.7 | 73.4 |

We noticed that even by training UOTA+MoCo-v2 from the very beginning (without resuming baseline), we can gain some improvement over the baseline model (73.0%). As we delay the resuming point, UOTA is improving until the resuming point is 20 epochs. If UOTA is activated beyond 20 epochs, the performance drops, as UOTA's effective training epochs reduces due to the constrained total training epochs (in total 200 epochs).

**X+UOTA Losses**. We explicitly describe the UOTA loss used for each baseline. This procedure can be briefly summarized as: we associate each $\bar{w}_{i,j}$ term (updated via Eq. (M.9), Eq. (M.11)) with the sample-wise loss of each baseline model, wherever the trainable feature $z_{i,j}$ is present. For "trainable", we mean the loss can be backpropagated through $z_{i,j}$. In this section, we expand Eq. (M.12) to include each specific SSL loss. We also use notation $z_{i,j}$ to replace the notation of trainable features in each original loss for illustration purpose (please see published baselines for original notation). For algorithms having multi-crop augmentations (i.e., MoCo-v2* and SwAV), we used in total 8 views for each baseline. This is equivalent to using some specific $M$ number of trainable sub-losses for each instance, according to pairwise optimization rule defined by each specific baselines. Here, for both BYOL and SwAV, we only illustrate the single sided **core** sub-loss used in each algorithm without showing its symmetric loss to avoid notation cluttering. This is because the symmetric loss is essentially the same as the original core sub-loss only having a different feature variable plugged in the loss. Note our implementation exactly resembles the symmetric loss under the swapping operation by computing $\bar{w}_{i,j}$ to associate with every loss having trainable features. Please see how the symmetric losses are constructed in [2] and [7].

1. MoCo-v2+UOTA

$$\mathcal{L}_{ours} = -\frac{1}{N} \sum_{j=1}^{M=1} \bar{w}_{i,j} \log \frac{\exp(z_{i,j}^T k^+/t)}{\exp(z_{i,j}^T k^+/t) + \sum_\ell^B \exp(z_{i,j}^T k_{i,\ell}^-/t)}, \quad (4)$$

2. MoCo-v2*+UOTA

$$\mathcal{L}_{ours} = -\frac{1}{MN} \sum_i^N \sum_j^M \bar{w}_{i,j} \log \frac{\exp(z_{i,j}^T k^+/t)}{\exp(z_{i,j}^T k^+/t) + \sum_\ell^B \exp(z_{i,j}^T k_{i,\ell}^-/t)}, \quad (5)$$

3. BYOL+UOTA

$$\mathcal{L}_{ours} = -\frac{1}{MN} \sum_i^N \sum_{j=1}^{M=1} \bar{w}_{i,j} \| z_{i,j} - z_{i,j}^\xi \|_2^2, \quad (6)$$

4. SwAV+UOTA

$$\mathcal{L}_{ours} = -\frac{1}{MNN_k} \sum_n^N \sum_j^M \bar{w}_{i,j} \left[ \sum_k^{N_k} q_{i,j}^{s,k} \log p_{i,j}^{t,k} \right], \quad (7)$$

where

$$p_{i,j}^{t,k} = \frac{\exp(z_{i,j}^T c_k/t)}{\sum_{k'} \exp(z_{i,j}^T c_{k'}/t)} \quad (8)$$

**Replacing $\mu_i$ by $z_i$ for MoCo-v2+UOTA.** When we implement UOTA on top of the baseline models with $M = 1$, $\mu_i$ becomes unavailable in the sense that there is only a single trainable feature $z_{i,j}$ during each training iteration. Take for instance, for the original MoCo-v2 baseline, there is only a single trainable query feature $z_{i,j} = q$, whereas $k^+$ is frozen during training. Correspondingly, to associate UOTA with MoCo-v2, we use the feature $z_i$ of original instance $x_i$ without any augmentation to replace $\mu_i$. In the main paper, this $\mu_i = z_i$ replacement operation has also been explained in Section M.3.3 and ablated in empirical study Section M.4.3 (Table M.3). The cost of this replacement though, is an extra forward of the original instance $x_i$.

**Learning rate for X+UOTA**. For each specific "X" model, we use the exactly same $lr$ as "X" under each specific optimizer to train each "X+UOTA" model.

**Augmentations for X+UOTA**. For all X+UOTA models, we use exactly the same augmentation policy as what the baseline model "X" published, without changing any hyperparameters.

**MoCo-v2**. For MoCo-v2 [5], we download code from the official website https://github.com/facebookresearch/moco. We train MoCo-v2 with the shuffleBN action defined in [5], and use an SGD optimizer with a learning rate $lr = 0.1$, momentum 0.9, weight decay $10^{-4}$. The learning rate is updated during training by following cosine decay rule [12].

**MoCo-v2\***. To implement MoCo-v2\*, we borrow training techniques from SwAV to facilitate the construction of a "multi-crop" variant of MoCo-v2. Specifically, the MoCo-v2\* loss is defined as:

$$\mathcal{L}_{MoCo} = -\frac{1}{MN} \sum_i^N \sum_j^M \log \frac{\exp(\boldsymbol{z}_{i,j}^T \boldsymbol{k}^+/t)}{\exp(\boldsymbol{z}_{i,j}^T \boldsymbol{k}^+/t) + \sum_\ell^B \exp(\boldsymbol{z}_{i,j}^T \boldsymbol{k}_{i,\ell}^-/t)}, \tag{9}$$

where $\boldsymbol{z}_{i,j}$ corresponds to the feature of the $j^{th}$ trainable view $\boldsymbol{x}_{i,j}$ of the $i^{th}$ instance, and resembles the $\boldsymbol{z}_{i,j}$ definition in our main file. The feature $\boldsymbol{z}_{i,j}$ here plays the role as query vector $\boldsymbol{q}$ defined as in [5]. To produce Table M.1, we use $M = 8$ for MoCo-v2\*, to facilitate comparison with the 8 views SwAV. Loss Eq. (S.9) is then only backpropagated through the trainable $\boldsymbol{z}_{i,j}$ during the training. Feature $\boldsymbol{k}^+$ and $\boldsymbol{k}^-$ are respectively the positive key and negative keys constructed following the implementation [5] without backpropagation. To train MoCo-v2\*, we firstly sample 8 augmentations under the multi-crop operation specified by SwAV, and we plug the feature $\boldsymbol{z}_{i,j}$ of each augmentation $\boldsymbol{x}_{i,j}$ into loss Eq. (S.9). To best leverage the multi-crop augmentation pipeline, we also use the exactly same training procedure proposed in SwAV (without using the SwAV loss) for MoCo-v2\*: we optimize Eq. (S.9) using the LARS [20] optimizer with a base learning rate of $lr = 2.0$, momentum of 0.9, weight decay of $10^{-6}$. According to SwAV, the learning rate policy follows a cosine learning rate decay schedule [12] with a "final learning rate" of 0.002.

**SwAV**. For the SwAV pretraining, we adapt implementation by referring to the published code https://github.com/facebookresearch/swav. We use the default training pipeline therein with parameters carefully tuned. These include LARS [20] optimizer with a base learning rate of $lr = 2.0$, momentum 0.9, weight decay of $10^{-6}$. The learning rate follows a cosine learning rate decay schedule [12] with a "final learning rate" of 0.002. Particularly, we tune for the number of epochs used in the SwAV "warm up" procedure. And we find using warm up 20 epochs, and 282 for the queue length and 300 for the number of prototypes are optimal on ImageNet100 dataset (corresponding to the SwAV scores reported in main paper). We believe we have tuned all of these hyperparameters to our best on the ImageNet100 dataset.

**BYOL**. In regard of the BYOL pretraining, we download the code from https://github.com/open-mmlab/OpenSelfSup and modify it so that the hyperparamers are suitable for pretraining on ImageNet100. We use the default LARS optimizer with a cosine decay learning rate schedule as described in [7]. We tune the base learning rate and find $lr = 0.6$ the best on Image100 dataset. To pretrain BYOL on ImageNet100, we follow the default definition on all of the remaining training hyperparameters as in [7].

**UOTA on supervised tasks.** We found that similar OOD issues raised by excessive distortions also exist in supervised learning. To show this, we download the code from [1], without altering any default augmentation parameters, and we train a supervised ResNet18 model on ImageNet100 for 100 epochs, referred to as baseline "Sup. IN100". We also train a model "Sup. IN100+UOTA" under the same setup. The top 1 accuracy for "Sup. IN100" is 81.9%, whereas the top 1 score for "Sup. IN100+UOTA" is 83.1%, verifying UOTA's effectiveness in such supervised learning scenario.

## 5.2 Linear Evaluation Protocol (Table M.1)

All the scores in Section M.4 are reported based on the Top-1 and Top-5 score via linear classification evaluation protocol. In detail, we firstly pretrain the networks using the baseline models described in Section S.5.1. We initialize the ResNet-18 network parameters with that copied from each pretrained model till the global pooling layer, then freeze the backbone parameters and only train the classifiers on the frozen features out of each pretrained network. For all training algorithm, we use a batch size $N = 256$ to train the classifier over 100 epochs.

For MoCo-v2 and MoCo-v2+UOTA linear classification training, we use SGD optimizer with $lr = 30$, without weight decay. The learning rate decays respectively at the $60^{th}$ and $80^{th}$ epoch. For MoCo-v2* and MoCo-v2*+UOTA linear classification training, we use SGD optimizer with $lr = 0.3$, without weight decay, and with a cosine learning rate schedule. For BYOL and BYOL+UOTA model, we use SGD optimizer with Nesterov momentum with $lr = 2.0$, without weight decay, and with a cosine learning rate schedule. For SwAV and SwAV+UOTA model, we use optimizer SGD, with $lr = 0.1$, weight decay $10^{-6}$, and with a cosine learning rate schedule.

## 5.3 Optimization for Pretraining other Outlier Removal Baselines (Table M.2)

In Table M.2, we present various outlier removal and noise robust algorithms when applied on the MoCo-v2* model. We now elaborate the optimization details when applying these approaches to SSL scenarios. In this section, we use $z_{i,j}$ to represent the trainable query feature $q$ in each approach. We firstly define the loss $f(z_{i,j})$ as:

$$f(z_{i,j}) = \frac{\exp(z_{i,j}^T k^+/t)}{\exp(z_{i,j}^T k^+/t) + \sum_{\ell}^B \exp(z_{i,j}^T k_{i,\ell}^-/t)}, \tag{10}$$

We then frequently refer to loss $f(z_{i,j})$ in the following section, in order to represent certain loss components in various SSL approaches. This helps ease the symbol cluttering issue.

**Focal**. We implement the Focal loss as follows.

$$\mathcal{L}_{Focal} = -\frac{1}{MN} \sum_i^N \sum_j^M \alpha(1 - f(z_{i,j}))^\gamma \log f(z_{i,j}), \tag{11}$$

where the hyperparameters of $\alpha = 1.0$ and $\gamma = 2.0$ offer the best performance.

**GCE**. We implement the GCE approach via loss:

$$\mathcal{L}_q(z_{i,j}) = \frac{1}{MN} \sum_i^N \sum_j^M \frac{(1 - f(z_{i,j})^q)}{q}, \tag{12}$$

$$\mathcal{L}_q(k) = \frac{1 - k^q}{q}, \tag{13}$$

$$\mathcal{L}_{GCE} = \begin{cases} \mathcal{L}_q(k) & if & f(z_{i,j}) \leq k \\ \mathcal{L}_q(z_{i,j}) & if & f(z_{i,j}) > k \end{cases} \tag{14}$$

where we tune the value of $k = 2.5 \times 10^{-4}$ and we find $q = 0.1$ offers the best performance.

**MIL-NCE**. We use exactly the MIL-NCE loss as published in [13], where we use 8 views ($M = 7$) formulation:

$$\mathcal{L}_{MIL-NCE} = -\frac{1}{N} \sum_i^N \log \frac{\sum_j^M \exp(z_{i,j}^T k^+/t)}{\sum_j^M \exp(z_{i,j}^T k^+/t) + \sum_j^M \sum_\ell^B \exp(z_{i,j}^T k_{i,\ell}^-/t)}. \tag{15}$$

**Debiased Contrastive Learning**. We modify the Debiased Contrastive Learning framework so that its multiple views originally used to approximate the expectation on positives are also suitable for multi-view MoCo-v2* framework. We refer to [6] and define:

$$Ng = max(-N\tau^+ \frac{1}{M} \sum_j^M \exp(z_{i,j}^T k^+/t) + \sum_\ell^B \exp(z_{i,j}^T k_{i,\ell}^-/t), N \exp^{-1/t}) \tag{16}$$

$$\mathcal{L}_{DCL} = -\frac{1}{MN} \sum_i^N \sum_j^M \log \frac{\exp(z_{i,j}^T k^+/t)}{\exp(z_{i,j}^T k^+/t) + Ng}, \tag{17}$$

We use the $t = 0.2$ and $\tau^+ = 0.01$ as suggested by [6] on ImageNet100.

For all the mentioned approaches above, we use $lr = 0.3$ and SGD optimizer to train the linear classifier on frozen features, as the same as MoCo-v2*. Then we report the Top-1 score in Table M.2.

# 6 Implementation Details of Section M.4.2 (Fig. M.2, Table M.3)

In Section M.4.2, we conduct ablation studies against hyperparameters crop-min and number of views. For producing Fig. M.2(a), we used $N_{warm} = 20$, $lr = 0.1$, to pretrain a ResNet18 with MoCo-v2+UOTA and MoCo-v2 respectively. The pretraining procedure and linear evaluation employs exactly the same protocols and same setups (and remaining hyperparamters) as described in Section S.5. For producing Fig. M.2(b), we used the same training schedule mentioned in Section S.5 for both MoCo-v2*+UOTA and MoCo-v2* baseline. We used $N_{warm} = 20$ for MoCo-v2*+UOTA. Please refer to Table S.3 for $\tau$ values. To produce Table M.3, we tune $\tau$ values under different estimate strategies as shown in Table S.4. The remaining setups for Table M.3 is the same as that for producing the MoCo-v2*+UOTA entry in Table M.2.

Table 3: Impact of different crop-min (1-6 columns) and various number of views (7-10 columns) for hyperparameter $\tau$ in UOTA.

| Hyperparameters | Crop-mini (MoCo-v2): | | | | | | Number of views (MoCo-v2*): | | | |
|---|---|---|---|---|---|---|---|---|---|---|
| | 0.10 | 0.12 | 0.14 | 0.16 | 0.18 | 0.20 | 4-view | 6-view | 8-view | 10-view |
| UOTA $\tau$ | 300 | 400 | 300 | 300 | 250 | 200 | 200 | 250 | 200 | 250 |

Table 4: Hyperparameter $\tau$ for different estimate strategies for Covariance and Mean in Eq. (M.10) of UOTA.

| Covariance & mean | $\mathbf{\Sigma} = \boldsymbol{I}$ & $\boldsymbol{\mu}_i$ | Local $\mathbf{\Sigma}_i$ & $\boldsymbol{\mu}_i$ | Global $\mathbf{\Sigma}$ & $\boldsymbol{z}_i$ | Global $\mathbf{\Sigma}$ & $\boldsymbol{\mu}_i$ |
|---|---|---|---|---|
| $\tau$ | 6 | 10 | 200 | 200 |

# 7 Implementation Details of Section M.4.2 (Table M.4, Table M.5)

To compare scores reported in NDA [15] as shown in our Table M.4, we run all the relevant baselines by strictly following the ResNet50 training setups of Table 6 in [15]. Specifically, for ImageNet-100 pretraining we have the following hyperhaprameters for MoCo-v2 and MoCo-v2+UOTA: batchsize $N = 128$, $lr = 0.015$, temperature $= 0.2$, feature dimentionality $= 128$. We implement unsupervised pretraining for 200 epochs and supervised training (linear classifier) for 100 epochs. For downstream linear classfication on ImageNet100, we use learning rate $lr = 30$.

To produce the binary classification result as reported in Table M.5 main file, we adopt the SGD optimizer with a learning rate $lr = 0.1$ momentum of 0.9 without weight decay for training 10 epochs. The learning rate is updated by cosine decay schedule.

# 8 Implementation Details of Section M.4.3 (Table M.4.6, Table M.4.7 )

In Section M.4.3, we evaluate the generalization capability of various algorithms by pretraining each ResNet50 Network on ImageNet1K. Here, we explain the training details of optimizing SwAV+UOTA loss. Specifically, we use all the default settings as published in [2] to train SwAV, i.e., with LARS optimizer, $lr = 0.6$, weight decay $10^{-6}$, and a cosine learning rate schedule with a final learning rate 0.006. We well reproduced the published score as in [2]. For all the data augmentations, we also follow exactly the augmentation policy and multi-crop operation as defined in SwAV. To train SwAV+UOTA loss, we find $lr = 1.2$ returns the best performance (corresponding to the score reported in Section M.4.3). For other hyperparameters, we set $\tau = 350$, $N_{warm} = 100$. Note, on ImageNet1K, our optimal learning rate for SwAV+UOTA model ($lr = 1.2$) doubles the optimal learning rate of SwAV ($lr = 0.6$), because we notice the loss on "effective number of samples" tends to have a larger impact on ImageNet1K (we did not tune learning rate on ImageNet100, and directly used model "X" learning rate to train "X+UOTA" on ImageNet100). We therefore correspondingly enlarge the learning rate, to make the effective loss scale of UOTA comparable to SwAV. This enlarged learning rate makes up for the loss of effective sample size (equivalently a constant multiplying the original UOTA loss) as defined in the conventional importance sampling technique.

To train the linear classification downstream task on ImageNet1K, we, again, adhere to SwAV's default setup to reproduce SwAV (SGD optimizer, weight decay $10^{-6}$, learning rate $lr = 0.3$ with

cosine schedule). For SwAV+UOTA, we use all the same training techniques and hyperparamters as in SwAV, except that we find $lr = 1.2$ is optimal on ImageNet1K given the new resultant SwAV+UOTA pretrained feature distribution.

In Section M.4.3, we also finetune the pretrained networks on the COCO dataset and evaluate the performance of various algorithms for object detection, instance segmentation and keypoint detection tasks. Our evaluation metrics are standard COCO AP (averaged over [0.5:0.95:0.05] $IoU$s), $AP_{50}$($IoU$=0.5) and $AP_{75}$($IoU$=0.75) scores respectively. For all of three downstream tasks, we attach FPN [11] to the corresponding training architectures. According to [8], we use extra batch normalization on the FPN layers, mask head and keypoint head. We perform the training on 4 V100 GPUs with total batch size $N = 16$, and we train all models for 90k iterations ($1\times$ schedule), according to [8]. All the hyperparameters not mentioned here follow the definitions in [8].