# OpenReview forum: "Improving Self-supervised Learning with Automated Unsupervised Outlier Arbitration"
_NeurIPS.cc/2021/Conference — NeurIPS 2021 Poster_

### Official Review · Reviewer_ifnT · 2021-07-04

**Rating:** 7
**Confidence:** 3

**Summary:**

The paper suggests that the different augmentations used as a positive pair during self-supervised training often lack the same semantic structure as the original image, and thus should be treated as an OOD sample. Using such OOD samples, they believe can hurt the downstream task performance. The aim should be to choose augmentations that offer the most data variance but are subject to the least semantic deviation. For this, they propose an importance sampling approach to weigh the loss for different augmentations differently. Empirically they obtain improvement for different downstream tasks across different datasets.

**Limitations And Societal Impact:**

Yes the authors have addressed their work's societal impact.

**Main Review:**

The idea of certain augmentations samples lacking semantic consistency and thus being weighed differently seems interesting. The authors do a good job of motivating the paper and showing results across different datasets and downstream tasks. The ablation studies are also nicely conducted.

My main doubts/concerns regarding the paper are the following -

1. While it is true that strong augmentations (such as RandomResizedCrop) often create OOD samples, but even for supervised settings people have been using such strong augmentations[1]. I believe the proposed approach can be extended even for such supervised settings. Can the reviewers show results over the supervised setting too, or otherwise comment on why their approach is not suitable for this direction.

2. Another work suggests explicitly generating OOD samples by augmentations such as Jigsaw, and then treats them as negatives during SSL training [2]. Can the authors compare their results to this paper?

3. The weights computed in the initial stages might not be useful, as the features extracted from the network are not well trained. What happens if you instead use a pre-trained network to determine the weights of the augmentations. Say for instance, you take a pre-trained MoCo-v2 and use the features extracted from this network to computed the weights of samples. How does this perform?


I will be willing to increase my score if the above points are addressed.

References -
[1] https://github.com/pytorch/examples/blob/master/imagenet/main.py
[2] Negative Data Augmentation. Sinha et al.

**Time Spent Reviewing:**

3

---

> ### Author Response · Authors · 2021-08-09
> **Response to Reviewer ifnT**
>
> Thanks very much for recognizing our technical contribution. Indeed, the sampling bias in SSL frameworks are worthy of fully examination. We hope UOTA can serve this goal by revealing the significance of potential OOD issues. We sincerely appreciate your constructive comments and we address your questions here.
>
>
>
> **A.1.** UOTA on supervised learning. We agree with your intuition: UOTA helps supervision as well, especially when the augmentation strength is large, or when there is noisy labels present in the training data. We here show that UOTA is effective on supervised learning tasks. Specifically, we download the code from [c], without changing any default augmentation parameters, and we train a plain ResNet18 in a fully supervised manner on ImageNet100 for 100 epochs (batchsize 256, weight decay 1E-4, cosine learning rate with base learning rate 0.1), which we referred to baseline model ''Sup. IN100''. We also train a model ''Sup. IN100+UOTA'' under the same optimization setup, to examine the effectiveness of UOTA on supervised tasks. The scores are (acc in %):
>
> | Model           | Top 1 | Top 5    |
> | --------------- | ----- | -------- |
> | Sup. IN100      | 81.9  | 95.0     |
> | Sup. IN100+UOTA | **83.1**  | **95.3** |
>
> The above experiments illustrate UOTA's advantage (1.2% gain) over baseline model even in an supervised learning setting. This result shows that strong augmentation can help, but excessively strong distortions will necessarily introduce noise and harm the pre-training, even for supervised learning. UOTA assumes that all of the augmentation factors (e.g., rotation, crop, etc.) lying in a subspace (via our modeling on global $\Sigma$ Eq. (10)), such that the sweet spot of all of these factors can be automatically (simultaneously) detected and balanced via the feature statistics. UOTA then consistently offers improvement gain for both unsupervised learning and supervised learning.
>
>
>
> However, we have to admit one point: the situation for supervised learning is a bit different, since the noise introduced by augmentation for supervised learning independently affects each training sample vs. its training label. In contrast, the supervision for self-supervised learning is usually pairwise. In other words, if any of the positively paired query or key turns out to be of different semantic classes, the self-supervision error would flow through each other, i.e., affecting both the query and key samples in a pairwise manner (via either direct backpropagation or momentum update), probably propagating the OOD error faster than in supervised learning.
>
>
>
> **A.2.** We are glad to see parallel works attending to the nature of OOD samples: they'd better be treated as negatives [a]. While UOTA tries to remove intrinsic OOD samples from positive, the work in the mentioned NDA paper [a] even considers manually generating OOD samples as negatives. These two methods differ in motivation, but both would be provably shown to result in reduced error bound on the downstream linear classification tasks, as per the discussion in paper [b]. To show this, we compare scores reported in NDA [a] and all the relevant baselines by strictly following the training setups of Table 6 in [a]. We consider 4 settings: 1) MoCo-v2 baseline (result published in [a]); 2) MoCo-v2 (ours) (result reproduced by us according to training details in [a]); 3) NDA +MoCo-v2 (results directly borrowed from paper [a]); 4) UOTA + MoCo-v2; The results are:
>
> | Model          | Top 1    | Top 5    |
> | -------------- | -------- | -------- |
> | MoCo-v2        | 69.4     | /        |
> | MoCo-v2 (ours) | 69.7     | 90.0     |
> | NDA+MoCo-v2    | 70.0     | /        |
> | UOTA+MoCo-v2   | **70.6** | **90.2** |
>
> As UOTA is orthogonal to NDA, we would even also try a fifth setting as 5) UOTA + NDA+MoCo-v2. But since the author did not respond to our query email asking for necessary training details, we unfortunately cannot provide this result at this time. But we would like to add this discussion in our updated version. At the end of the day, when purely comparing NDA against UOTA, we find UOTA provides relatively better performance under the specific training scenarios in [a]. We hope these experimental designs serve the goal to clarify your questions.
>
>
>
> **A.3.** We agree with your intuition that random initialization is bad. That's why we were using resuming strategy when implementing UOTA. Take for example, in main paper Table 1, the model MoCo-v2+UOTA is initialized by resuming from MoCo-v2's 30 training epochs, and we train MoCo-v2+UOTA for the remaining 170 epochs (in total 200 epochs for fair comparison). For other X+UOTA formulations, please refer to supplementary material's Table 1. Here, we would like to further ablate the effect of the resuming point (from which epoch to resume a MoCo-v2 baseline model by adding UOTA, while the total training epochs remains as 200 epochs on ImageNet100 dataset. The reference accuracy of a baseline MoCo-v2 is 73.0%.
>
> | Resume point | 0    | 10   | 20       | 30   | 50   | 100  | 150  |
> | ------------ | ---- | ---- | -------- | ---- | ---- | ---- | ---- |
> | Top 1        | 73.5 | 73.7 | **74.0** | 73.9 | 73.6 | 73.7 | 73.4 |
>
> We noticed that even by training UOTA+MoCo-v2 from the very beginning (without resuming baseline), we can gain some improvement over the baseline model (73.0%). As we delay the resuming point, UOTA keeps improving till the resuming point =20​ epochs. If UOTA is activated beyond 20 epochs, the performance drops, as UOTA's effective training epochs reduces due to the constrained total training epochs (in total 200 epochs).
>
> Thanks again for your comments!
>
> [a] A. Sinha, et al., ''Negative data augmentation''. ICLR 2021.
>
> [b] S. Arora, et al., ''A theoretical analysis of contrastive unsupervised representation learning'', ICML 2019.
>
> [c] ImageNet training in PyTorch: https://github.com/pytorch/examples/tree/master/imagenet

---

> > ### Comment · Reviewer_ifnT · 2021-08-15
> > **Follow up**
> >
> > I thank the authors for addressing all my above-mentioned concerns. I would encourage them to include all the above 3 experiments in their paper as these seem to strengthen the paper.
> > I am increasing my rating accordingly.

---

> > > ### Author Response · Authors · 2021-08-16
> > > **Thank you for your review comments!**
> > >
> > > Many thanks again for your careful reviews and constructive comments! We are glad that our former response has clarified the questions, and thanks for increasing the score. It has been our pleasure to try out the suggested experiments. Surely, we will add the above empirical results in our updated version of paper.

---

### Official Review · Reviewer_iGgP · 2021-07-15

**Rating:** 6
**Confidence:** 3

**Summary:**

This contribution is based on two main claims: i- using data augmentation to create different views from an image might result in out-of-distribution instances; and ii- employing such out-of-distribution views as positive instances in self-supervised approaches that rely on comparing representations from positive and negative views of an example, might be harmful for downstream performance. To address these issues, the authors proposed to apply an out-of-distribution detection approach to estimate how likely it is for a positive view to be out-of-distribution and use this information to weight the contribution of each positive pair when computing the learning criterion. The proposed strategy, named Unsupervised OuTlier Arbitration (UOTA), is then incorporated to three previously proposed self-supervised learning approaches and empirically validated on ImageNet and MS-COCO. Comparisons with other out-of-distribution detection techniques were also provided. Overall, UOTA showed to marginally improve the downstream performance of self-supervised pretraining both with respect to the compared detection approaches as well as self-supervised models without any mechanism to compensate for potential out-of-distribution positive views.

**Limitations And Societal Impact:**

Yes.

**Main Review:**

The main positive aspect of this work is to raise a discussion on using out-of-distribution detection as a tool for improving self-supervised methods that rely on contrastive learning between multiple views. I also appreciated the large number of experiments and ablation studies presented in the empirical analysis. In spite of that, I think this contribution is not ready for publication, mostly due to a lack of clear evidence to support its main claims and conclusions. In addition to that, I found the manuscript quite hard to follow (it was necessary to read some sections a few times to grasp at least their main content). In the following, I further detail my concerns and provide suggestions to address them.


- Unsupported claims:

  - The authors claim in line 30 "We find these produced views even semantically deviate from the original instance and thus behave like out of distribution (OOD) samples" and in line 92  "we reveal that OOD noise is inherently existing in positive view sampling pipelines for a broad range of mainstream SSL approaches". However, it is not clear to me which part (if any) of the discussion presented in Sections 3.1, 3.2, and 3.3 supports this claim. As far as I understood, the example provided in Figure 1 is the only evidence showing that augmentation approaches for generating positive views are introducing out-of-distribution instances. To address this issue in the current version of the manuscript, experiments such as showing whether linear classifiers trained on top of features extracted by self-supervised pretrained models are able to distinguish augmented and original examples could be interesting (i.e. estimating the $\mathcal{H}$-divergence [1] between augmented and original data distributions).

  - Moreover, in line 32:  "The model therefore fail to generalize in some certain parameter region due to the interference of the OOD samples". I also couldn't find in the manuscript no piece of evidence to support such claim. How exactly out-of-distribution views are hurting generalization performance? Aren't there cases where enforcing the model to match positive views that might be considered as coming from different distributions could actually be helping the encoder to discard irrelevant information for the downstream the task?

  - Line 369: "Empirical study well corroborates our hypothesis and shows strong advantage of our UOTA approach over the state-of-the-art SSL approaches." The results are not presented along with error bars or any statistical significance analysis, therefore, I don't think it is possible to conclude from the empirical evaluation that UOTA shows a *strong* advantage with respect to previous approaches. In Table 4, for example, the improvement obtained by SwAV+UOTA was less than 1% in comparison to SwAV when training both models during 200 epochs, which might be due to other factors rather than due to the use of UOTA. To further illustrate my point, in Figure 3b of [3], it is shown that after training SwAV for 200 epochs it reaches 73.9% top-1 accuracy on ImageNet1k, surpassing SwAV+UOTA with the same budget of epochs. To address this issue, I suggest the authors to make it clear in the manuscript that the experiments do not indicate that UOTA provides a *strong* advantage with respect to the considered baselines.


- Questions and other concerns:
  -  Surprisingly, the introduction contains no reference to support several statements related to previous work. For example, "So far, the most prevailing and effective assumption, is to force views from the same instance to be invariant in the feature space." (line 25) need to be followed by the corresponding references.

  - It seems to me that different augmentations will shift the data distribution in different ways and this aspect should also be discussed in the manuscript. More specifically, I'm looking forward to understanding whether it would be helpful to only consider in the pipeline augmentations that generate small shifts. In case only these augmentations are used to generate positive views, how would this affect the downstream performance of the self-supervised model, *i.e.* what is the interplay between the distribution shift induced by an augmentation pipeline and the generalization capability of the self-supervised model on downstream tasks?

  - Although I understand that contrastive learning approaches are very common for developing self-supervised methods, I believe the authors should also acknowledge in the manuscript the related literature which is based on different strategies, such as clustering. Please cite, discuss, and include for comparison in the experiments approaches such as DeepClusterv2 [2, 3] (it showed strong performance on several experimental settings, and pretrained models along with the original implementation are available on github).

  - Would the proposed importance sampling approach be also useful to weight the importance of augmentations on purely supervised tasks?

  - Line 109: specify that $\theta$ corresponds to the model's parameters.

  - Line 126: what is an M-estimator?

   - Line 180 "A radical sampling distribution on n can simply contradict the desired prerequisite hypothesis (invariance) at hand and increases bias". It is not clear to me what radical sampling means. Please clarify.

- Minor (not affected the final score):
  - Typo in line 277: "raining" -> training
  - Line 374: "appropriate" -> inappropriate?


[1] Ben-David, Shai, et al. "A theory of learning from different domains", 2010.

[2] Caron, Mathilde, et al. "Deep clustering for unsupervised learning of visual features", 2018.

[3] Caron, Mathilde, et al. "Unsupervised learning of visual features by contrasting cluster assignments", 2020.

**Time Spent Reviewing:**

6

---

> ### Author Response · Authors · 2021-08-09
> **Response to Reviewer iGgP (Part 3/3)**
>
> - Questions and other concerns. (continued from Part 2/3)
>
>   - **A.4.** Thanks for spotting the reference issue. We apologize for this, and we would like to associate each claim in the introduction with appropriate references such as [a] [b] [c].
>
>   - **A.5.** We are happy to discuss OOD samples' impact on the downstream tasks. Firstly, the short answer to your question ''can we achieve good result if we simply use the augmentation with small shift'' is: no.  The longer answer is: no, because we are sacrificing data variance, and we do not know how small is good enough for each instance. Interestingly, this question seems to be the point UOTA trying to address throughout the paper: we do want large data variance through strong augmentations, but we should be careful not to introduce noise through extremely large data distortions. Perhaps an illustrative example is the extreme case when the variance is infinity. If this happens, the features belonging to different classes overlap, the estimator will lose motivation to achieve any desired semantic invariance. As the augmentations are applied through multiple dimensions: e.g., crop size, lighting condition, rotation and etc., UOTA assumes that all of these factors lying in a subspace (via our modeling on global $\Sigma$ Eq. (10)), such that the sweet spot of all of these factors can be automatically (simultaneously) detected and balanced via the feature statistics. Based on those statistics, UOTA reassures us that the estimator bias is best reduced without scarifying too much data variance we desired.
>
>     We also refer Reviewer iGgP to paper [e], which theoretically discuses the generalization ability of contrasive pre-training approaches on downstream tasks. In [e], authors clearly show that, if negative sample is from the same semantic class with the positive, the upperbound of error on downstream linear classification is provably guaranteed to increase [e]. Similarly, it can be concluded from [e] (Lemma 4.3) via mild modification that, if a pair of positive samples are not from the same semantic class (e.g., OOD samples showing different semantics or strong deviations from the semantic, such as bunny crop vs. straw crop in our paper), the upper bound of linear classification error in the downstream tasks will also be guaranteed to increase, leading to provable worse downstream performance.
>
>   - **A.6.** Comparisons with cluster based approach. We have included SwAV [c] as an important baseline throughout the paper already, for we believe SwAV is right a typical clustering based approach. Reviewer iGgP also refers to deepclusterv2 [g]. We are happy to compare with deepclusterv2. We found in the SwAV paper that DeepCluster2 provides similar performance of SwAV (Figure 3 in [c]) though. But we will certainly refer to github, reproducing deepcluster2 scores and add this comparison in our updated version. But again, UOTA is orthogonal to deepcluster2, and can be potentially added on top of deepcluster2, too.
>
>   - **A.7.** UOTA on supervised tasks. Yes, we believe UOTA also benefits supervised learning on OOD issues, with mild difference though. We here firstly show that UOTA is empirically effective on supervised learning tasks. Specifically, we download the code from [d], without changing any default augmentation parameters, and we train a plain supervised ResNet18 model on ImageNet100 for 100 epochs (batchsize 256, weight decay 1E-4, cosine learning rate with base learning rate 0.1), which we referred to baseline model ''Sup. IN100''. We also train a model ''Sup. IN100+UOTA'' under the same optimization setup, to examine the effectiveness of UOTA on supervised tasks. The scores are (acc in %):
>
>     | Model           | Top 1    | Top 5    |
>     | --------------- | -------- | -------- |
>     | Sup. IN100      | 81.9     | 95.0     |
>     | Sup. IN100+UOTA | **83.1** | **95.3** |
>
>     The above results corroborate that UOTA is also efficient on supervised learning. However, we have to admit one point: the situation for supervised learning is a bit different, since the noise introduced by augmentation for supervised learning independently affects each training sample vs. its training label. In contrast, the supervision for self-supervised learning is usually pairwise. In other words, if any of the positively paired samples are likely from different semantic classes (due to large distortion), the self-supervision error would flow through each other in a pairwise manner (via either direct backpropagation or momentum update), probably propagating the OOD error faster than in supervised learning.
>
>   - **A.8.** M-estimator is an estimator where the objective function is a sample average, and the goal is to optimize such sample average based objective function w.r.t. estimator parameters. If interested, please also refer to Wikipedia or [f] for more detailed discussion on properties of M-estimator.
>
>   - **A.9.** ''radical sampling''. For ''radical'', we mean an extremely large distortion applied on images, which causes drastic feature deviation from its original class semantics. Also see answer A.4. on this. We can certainly modify this wording in the updated version, if requested.
>
>   - **A.10.** Thanks for catching the typos, we promise to modify the wordings, presentations and story flows wherever necessary.  Thanks again for your comments!
>
>
>
> [a] X. Chen, et al., ''Exploring Simple Siamese Representation Learning'', CVPR 2021.
>
> [b] X. Chen, et al., ''An Empirical Study of Training Self-Supervised Vision Transformers'', arXiv, 2020.
>
> [c] M. Caron, et al., ''Unsupervised Learning of Visual Features by Contrasting Cluster Assignments'', NeurIPS 2020.
>
> [d] ImageNet training in PyTorch: https://github.com/pytorch/examples/tree/master/imagenet
>
> [e] S. Arora, et al., ''A Theoretical Analysis of Contrastive Unsupervised Representation Learning'', ICML 2019.
>
> [f] Huber, P.J., Robust statistics (Vol. 523). John Wiley & Sons. 2004.
>
> [g] M. Caron, et al., ''Deep Clustering for Unsupervised Learning of Visual Features'', ECCV 2018.
>
> [h] J. Grill, et al., ''Bootstrap Your Own Latent: A New Approach to Self-Supervised Learning'', NeurIPS, 2020.

---

> > ### Comment · Reviewer_iGgP · 2021-08-27
> > **Updating my score**
> >
> > Dear authors,
> >
> > Thank you for your careful response to my comments.
> >
> > I believe the majority of my concerns were addressed by the rebuttal, especially the ones regarding the "unsupported claims" part of my review. Thus, I increased my score accordingly.
> >
> > As a final remark, I would like to point out that, as opposed to what was mentioned in the part 2/3 of the response, my review does not contain the following statement ''improvement less than 1​% on ImageNet1K is weak and that such improvement might be due to other factors rather than due to the use of UOTA''. Notice that, as I mentioned in my review, my actual concern is about the statistical significance of the reported results rather than absolute improvement.
> > Luckily, my concern was partially addressed by the following statement in the rebuttal: "To be frankly, for all experiments presented in Table 1,2,3, we have repeated each experiment for 5 times with different random seeds, by reporting the mean value in the main paper. The accuracy variance is below 0.02%."
> > As it seems the authors misunderstood my point, I would like to further remark that, given the limited amount of time for running new experiments, including the variance values to the respective tables would be sufficient to address my concern.
> > At this point, it is not clear to me that whether the authors plan to add these values to an updated version of the manuscript or not (I couldn't find it in the supplementary material as well).

---

> > > ### Author Response · Authors · 2021-08-27
> > > **Thanks for your feedback!**
> > >
> > > We sincerely appreciate your constructive review comments and kind suggestions. Many thanks for increasing the score. We are happy that our response has addressed your concerns, and we felt inspired to try out the suggested experiments. For sure, we promise to add the suggested statistical significance analysis in the updated version of the main paper. Specifically, we will add error bars (variance values) to associate with each reported result in the empirical study section of our main paper, in order to abate the impact of randomness via statistical significance evaluations. We would also include other suggested empirical study (e.g., the binary OOD classification experiment) into the updated version of the main paper (or in the supplementary material given limited space). We believe these revisions may further improve our paper. Thanks!

---

> ### Author Response · Authors · 2021-08-09
> **Response to Reviewer iGgP (Part 2/3)**
>
> - (Unsupported claims): continued from Part 1/3
>   - **A.3.** Empirical Results:
>
>     - Firstly, we are surprised by Reviewer iGgP's judgment that ''improvement less than 1​% on ImageNet1K is weak and that such improvement might be due to other factors rather than due to the use of UOTA''. In fact, we have taken the fairness issue very seriously when conducting all experiments and comparisons. We have also strictly followed the conventions (e.g., tasks, dataset, assessment metric) in the SSL community when empirically evaluating all of the models. Particularly, in order to ablate the advantage of UOTA over SwAV, we have reproduced the SwAV pre-training performance as faithfully as we can (and we managed to reproduce the SwAV published result). We carefully make sure all relevant pre-training optimization details (including augmentation policy, weight decay, batchsize, network architecture, parameter number and all related parameters) of UOTA to follow exactly the same policy as the baseline SwAV, under exactly the same experimental conditions. Still, we have no clues what the mentioned ''might be due to other factors rather than due to the use of UOTA'' is referring to. Please note, UOTA's performance closely adheres to its baseline model, since the computation of the Radon Nikodym derivative (defined in line 159, which boil down to Eq. (9) in paper) is based on the statistics provided by the baseline model. If there were any other factors affecting the baseline's performance, UOTA would also correspondingly change, due to the change of feature statistics.
>
>     - Secondly, we may not accept the criticism that ''UOTA is not improving the baseline by performance gain less than 1%​''. We kindly refer the reviewer to SimSiam [a], in which Table 4 offers reference reporting recent state-of-the-art SSL performance. For all the models pre-trained for 200 epoch, we note BYOL (70.6%) improves over MoCo v2 (69.9%) by 0.7%, and SwAV (69.1%) even drops below MoCo v2 by 0.8% when the multi-crop is removed from SwAV. Regarding SimCLR, the more recent MoCo-v2 is 1.6% better, which relies on doubled network parameter size (to produce query and key) and large memory banks though. Given those references, we insist that UOTA+SwAV's improvement 0.8% over SwAV baseline is actually effective. We think this 0.8% improvement effectively raises a warning flag to the community, and it verifies that OOD samples has a clear and non-negligible impact that should be attended to. If we further refer to MoCo v3 [b] page 3, Section 4.1, X. Chen et al. observed that, even a 0.1%-0.3​% performance drop can be indicative of instability issues for SSL methods. To be frankly, for all experiments presented in Table 1,2,3, we have repeated each experiment for 5 times with different random seeds, by reporting the mean value in the main paper. The accuracy variance is below 0.02%. For the ImageNet1K pre-training, we implemented the UOTA pre-training in 2 runs, leading to almost the same results (73.51% vs. 73.53%) on linear classification. Also, we observe that for the linear classification task on ImageNet1K, UOTA performs better than SwAV baseline in ​703​ classes among total 1000 classes, showing statistical significance of our improvement in most classes.
>
>     - Thirdly, Reviewer iGgP refers to 73.9% reported in Figure 3(b) in SwAV paper [c]. According to SwAV paper [c] (Section 5.3), 73.9​% corresponds to the SwAV model pre-trained with batchsize 4096, whereas UOTA reported in our paper is strictly based on SwAV baseline pre-trained with only batchsize 256. While SwAV itself recognizes the sensitivity on batchsize in [c] (since it is a cluster based algorithm), we are hesitant whether this comparison under different training conditions is fair. Please note, UOTA+X 's performance is closely related to its baseline model X since the computation of the Radon Nikodym derivative hinges on the statistics from the baseline model. If there were any other factors improving the baseline model's performance (such as batchsize), UOTA would also correspondingly improve.
>
>       Also, to reassure Reviewer iGgP that UOTA's leading performance is consistent across tasks, and that UOTA indeed improves the network generalization ability instead of being lucky, we showcase additional linear classification performance in the following transfer learning (by using the same pre-trained model on ImageNet1K in main paper). The tasks are defined in BYOL [h]. The accuracy is top1 score reported in %:
>
>       | Model | Aircraft | Birdsnap | Caltech101 |   Cars   | CIFAR10  | CIFAR100 |   DTD    | Flowers  |   Food   |   Pets   |  SUN397  |   VOC    | Average  |
>       | ----- | :------: | :------: | :--------: | :------: | :------: | :------: | :------: | :------: | :------: | :------: | :------: | :------: | :------: |
>       | Sup.  |   42.6   | **57.2** |    90.9    |   62.8   |   90.0   |   73.4   |   68.8   | **89.7** |   71.3   | **92.4** |   60.3   |   87.2   |   73.9   |
>       | SwAV  |   45.5   |   49.0   |    98.0    |   62.2   |   90.8   |   73.5   |   72.2   |   88.9   |   74.3   |   87.1   |   64.4   |   88.1   |   74.5   |
>       | UOTA  | **46.3** |   50.6   |  **98.3**  | **63.5** | **91.4** | **74.8** | **73.6** |   89.3   | **76.1** |   87.7   | **65.4** | **88.4** | **75.5** |
>
>       *Sup. refer to supervised model as a baseline reference as described in [d]. *SwAV refers to [c]. *UOTA is the UOTA+SwAV model, the same pre-trained model for producing Table 4 and 5 in our main paper.
>
>       We also challenge the UOTA model on the semi-supervised learning tasks, as defined in SwAV [c] paper.
>
>       | Model     | 1% labels (top1) | 1% labels (top5) | 10% labels (top1) | 10% labels (top5) |
>       | --------- | :--------------: | :--------------: | :---------------: | :---------------: |
>       | SwAV      |       49.6       |       76.1       |       67.7        |       88.8        |
>       | UOTA+SwAV |     **51.3**     |     **77.4**     |     **68.7**      |     **89.2**      |
>
>     (**To be continued in Part 3/3.**)

---

> ### Author Response · Authors · 2021-08-09
> **Response to Reviewer iGgP (Part 1/3)**
>
> We sincerely appreciate your constructive suggestions and we take these reviews seriously. Admittedly, there might be some missing reference, leaving some part of our technical claims ambiguous. We apologize for those and would try our best to disambiguate in this response. But we are also very keen to clarify some key misunderstanding here, especially regarding your comments on ''evidence''. **All references are specified in the Part 3/3 response**.
>
> - Unsupported claims:
>   - **A.1.** ''We reveal that OOD noise is inherently existing in positive view sampling pipelines for a broad range of mainstream SSL approaches''. Actually, the whole empirical study section on ImageNet100 and Imagenet1K serves as a good empirical justification to this claim: i.e., OOD sample matters. Excessively strong augmentation will distort the feature distribution and introduce OOD samples. Please refer to line 268-273: i.e., by taking into account the impact of OOD noise, downgrading the influence of OOD samples alone is consistently offering much better performance (via UOTA) than ignoring the OOD noise. We believe those empirical evidences have well ablated the net effect of OOD noise on various SSL approaches, and clearly reflected the detrimental influence of OOD noise, echoing our claim you concerned: OOD noise indeed matters for these SSL baselines.
>
>     Also, we appreciate reviewer iGgP's suggestion on a 2 class linear classification between OOD samples and clean samples. As requested, we show that OOD samples and original image are indeed linear separable on frozen features. To achieve this goal, we need some supervision based on the computed Radon Nikodym derivative ($w_{i,j}$ values according to Eq. (9) and Eq. (11)). In detail, we choose ImageNet100 training set (126,689 images) as training set and ImageNet1K val set (50,000 images) as test set. Each of the training data is randomly augmented into 5 views. We then reorder all the samples according to their computed $w_{i,j}$ value in an ascending order. We then sequentially split such reordered augmentation set into 5 groups with equal sample size. Different groups then reflect the detected semantic uncertainty via UOTA. In this way, Group 1 contains the lowest $w_{i,j}$ valued samples, showing strongest sign of OOD sample, while the Group 5 has the highest $w_{i,j}$ samples, most resembling the semantic of the origional training sample. Note, the extra Group 6 is a baseline reference: we randomly divide the clean training set into two subsets, with each subset representing class 1 and class 0. The test data are augmented in the same way and split into 5 groups, too. We then train the binary classifier on top of the frozen pre-trained model (SwAV+UOTA model producing Table 4 in paper), by respectively treating each Group n as class 1 (OOD class), with clean data seen as class 0 (class 0 and 1 have equal number of training data). We adopt the SGD optimizer with a learning rate lr=0.1​, momentum of 0.9​ without weight decay for 10 epochs. The learning rate is updated by cosine decay schedule. We report such 2 class linear classification training and test (test is from the group indexed identically with the corresponding training group) accuracy follows:
>
>     | OOD sample group | Group 1 (lowest) | Group 2 | Group 3 | Group 4 | Group 5 (highest) | Group 6 (Non-ood) |
>     | ---------------- | ---------------- | ------- | ------- | ------- | ----------------- | ----------------- |
>     | Training acc     | 99.9             | 97.0    | 92.2    | 87.9    | 86.5              | 51.5              |
>     | Test acc         | 99.8             | 96.3    | 91.1    | 86.7    | 84.9              | 49.7              |
>
>     As can be seen, no matter from which group the OOD samples are from, the linear classifier on frozen features always show clear statistical significance against the reference baseline Group 6. Notably, as the data exhibits weaker OOD signs (e.g., Group 5), the training becomes harder and harder, returning lower training accuracy, while the OOD samples still demonstrate much stronger linear separability against the baseline model Group 6.
>
>   - **A.2.** Section 3 is not intended to cover how much each baseline suffers from OOD samples. Rather, Section 3 analyzes why OOD is a potential issue for all baselines. To reach this purpose, Section 3 non-trivially decomposes the estimation error into several components, with the goal to reveal the hidden estimator bias-variance trade-off existing behind estimator error: SSL methods certainly need data variance, but it is equally important to constrain such strength. The error term in Eq. (5)(6) due to excessively large distortion would increase the estimator bias, and thus increase the total estimation error, as per the discussion in Lemma 1. But if the data variance is too small, the benefit of error reduction via Eq. (8) will be lost due to insufficient data variance. Section 3 then constitute useful avenues for advancing our conceptual motivation and derivation of UOTA to trade off the augmentation strength. Most importantly, by eliminating the OOD noise (via UOTA like approaches), we are able to conveniently balance such hidden variance-bias trade-off, and to improve the downstream tasks, which also proves our hypothesis that OOD noise matters. Please see response A.5 in part 3/3 for further discussions on generalization ability. (**To be continued in Part 2/3.**)

---

### Official Review · Reviewer_wYZX · 2021-07-16

**Rating:** 5
**Confidence:** 5

**Summary:**

This paper presents an algorithm named UOTA for self-supervised representation learning. The method is based on the observation that some low-quality negative samples are generated during the learning procedure and these bad samples can confuse the deep networks. So, a Gaussian distribution is built to estimate the weight of each sample and added to the loss term. Experiments show consistent accuracy gain.

**Limitations And Societal Impact:**

Yes

**Main Review:**

This paper presents an algorithm named UOTA for self-supervised representation learning. The method is based on the observation that some low-quality positive samples are generated during the learning procedure and these bad samples can confuse the deep networks. So, a Gaussian distribution is built to estimate the weight of each sample and added to the loss term. Experiments show consistent accuracy gain.

The major concern of this paper lies in the technical contribution. Though some ablations are made to validate the effectiveness of introducing a Gaussian distribution to estimate the quality of positive samples, this is actually a small add-on to the existing self-supervised learning approaches. I buy the motivation, the method, and the mathematical analysis, but I just think that the contribution does not achieve the standard of NeurIPS.

To add more words, I do think that the current self-supervised learning methods rely too much on the negative samples, which is not good. However, I really expect some efforts of alleviating this issue by (1) finding key objects in the image so that the positive samples can largely preserve the quality of these key objects; or (2) finding a new proxy that bypasses the requirement of positive or negative pairs. I know that it is unfair to ask the authors to follow the suggested path, yet I just think that adding a weight to each positive sample forms a limited contribution, so I cannot suggested acceptance for now.

BTW, there are some efforts in adding stronger augmentations to achieve better performance of self-supervised learning. Do you have any comments of applying the proposed approach, UOTA, to these stronger augmented samples?
[A] Wang et al., Contrastive Learning with Stronger Augmentations, 2020.
[B] Xu et al., Seed the Views: Hierarchical Semantic Alignment for Contrastive Representation Learning, 2020.

**Time Spent Reviewing:**

1

---

> ### Author Response · Authors · 2021-08-09
> **Response to Reviewer wYZX**
>
> Many thanks for your comments. We appreciate that you liked our motivation, method and mathematical analysis, whereas you still felt hesitant to accept our idea for NeurIPS. We sincerely hope this response letter can help relieve your concerns.
>
> Since your comments do not point to any technique details in our paper, we would like to clarify some misunderstanding. Firstly, UOTA is not about ''adding weight''. The Radon Nikodym derivative (your mentioned weight) obtained via Eq. (9) is the direct result that naturally arises from our analysis Lemma 1. In other words, sampling distribution on SSL positives needs to be corrected via techniques such as importance sampling, if evidence shows that OOD sample is present. ''Weighting'' the samples is never the motivation of our paper. Instead, it is one of the effective means guaranteed to correct for the biased positive sampling distribution. In fact, we believe UOTA is the first work attempting to investigate such OOD issue behind mainstream SSL methods and the associated positive sampling bias problem. Section 3 then constitutes useful avenues for advancing our conceptual motivation and derivation of UOTA to trade-off the augmentation strength. UOTA provides both theoretical accessible and practicality useful solution to this issue. Notably, we have derived an entirely new and non-trivial analysis of recognizing ''positive sampling variance'' as a double-edged sword, whose strength can be balanced and adjusted in an unsupervised way. UOTA elegantly achieves this goal via reducing the expected estimator error through Lemma 1 and Theorem 1, by resorting to the simple feature statistics (without incurring much complexity).
>
> Instead of claiming ''weighting helps'', our empirical observation and theoretical derivation actually aims at an alternative goal: SSL methods based on pairwise measurements indeed do favor sampling variance (some form of strong augmentation), but such variance (strength of augmentation) should be properly constrained as per our analysis, such that OOD samples and excessive distortions on semantics are best avoided. Our empirical results justify that, training with unsupervised OOD removal schemes such as UOTA can consistently improve downstream tasks performance, corroborating our claims in Theorem 1. Otherwise, the OOD samples (samples with excessive strong distortions) will eventually become a disadvantage that even sabotages the generalization capability of the SSL pre-training model (through increasing the expected estimator error). The above arguments also answer your question regarding the two mentioned references [a] Wang et al and [b] Xu et al - We emphasize that UOTA never denies that ''augmentation'' helps. Rather, UOTA warns against risks through excessively strong distortions. If we look at the papers you mentioned [a] [b], we find their motivation is actually not contradicting ours. The abstract of [a] even clearly states that: ''the strong augmentations distorted the image structures, resulting in difficult retrieval '', agreeing with our observations that stronger augmentation does not always help. To illustrate this, we still refer to your mentioned [a], and tune the strength of augmentation on ImageNet100 according to [a]. We deploy the MoCo-v2 baseline as described in main paper Table 1. We then repeat the strong augmentation on each image for M times (larger M, stronger augmentation), suggested by [a]. We train each MoCo-v2 or MoCo-v2+UOTA model with different augmentation strengths and report the accuracy in %:
>
> | M              |   0    |   1    |   5    |   10   |
> | -------------- | :----: | :----: | :----: | :----: |
> | MoCo-v2        |  73.0  |  75.3  |  76.0  |  72.8  |
> | MoCo-v2 + UOTA |  74.0  |  76.2  |  77.2  |  74.6  |
>
>
> We observe that MoCo-v2 keeps improving until augmentation strength increases to ''5'', but MoCo then suffers drastically when augmentation strength is beyond ''5''. Nevertheless, UOTA always is able to correct for the sampling distribution, gaining improvement over the baseline. Strength M=10 particularly manifests UOTA's advantage. These experiments strongly support our hypothesis that strong augmentation does not always help, but UOTA can adaptively relieve the harm.
>
>
>
> It seems Reviewer wYZX shows prejudice against the term ''weighting'', and determines ''weighting'' is a ''small add on''. We may not agree. Many ground breaking works in the machine learning community show optimization equivalence to some form of reweighting, although weighting itself is not their motivation at all. For example, the classical expectation maximization (EM) itself is iterative reweighting scheme, that the latent variable is reweighted by the posterior estimation during each iteration; The conditional entropy regularizer $p\log p$ [c] often used to ensure cluster assumption on class boundaries, where $p$ essentially reweighs each samplewise cross entropy $y\log p$; Self-paced learning [d] and Generalized cross entropy [e] are typical methods where the update rules are reweighting samples, whereas the goal was to solve for the auxiliary variables of the original optimization problem. The generative model beta-VAE [f] simply scales the KL term by $\beta$ w.r.t. the conventional VAE, theoretically guaranteed to improve the performance. Even if in the SSL community, take the recent published [g] for example, the formulation therein essentially is reweighting the terms in the denominator. Many of the above papers are published at NeurIPS, or prestigious machine learning proceedings with hundreds or thousands of citations. We therefore kindly invite Reviewer wYZX to reconsider assessment on UOTA's essential technical contributions, depending on the above discussions.
>
>
>
> The comments ''current SSL methods rely too much on the negative samples, which is not good'' is beyond the discussion scope of our paper. However, we showed that UOTA was applicable to such SSL baselines completely free of negatives, and we believe many emerging state-of-the-art SSL methods are free of negative samples [h], or have shown theoretical correctiveness of such frameworks [i]. Thanks again for your comments!
>
>
>
> [a] X. Wang, et al., ''Contrastive learning with stronger augmentations'', arXiv 2020.
>
> [b] H. Xu, et al., ''Seed the views: Hierarchival semantic alignment for contrastive representation learning'', arXiv 2020.
>
> [c] O. Chapelle, et al., ''Semi-Supervised Classification by Low Density Separation'', AISTATS, 2005.
>
> [d] M. Kumar, et al., ''Self-Paced Learning for Latent Variable Models'', NIPS 2010.
>
> [e] Z. Zhang, et al., ''Generalized Cross Entropy Loss for Training Deep Neural Networks with Noisy Labels'', NeurIPS 2018.
>
> [f] I. Higgins, et al., ''beta-VAE: Learning Basic Visual Concepts with a Constrained Variational Framework'', ICLR 2017.
>
> [g] C. Chuang, et al., ''Debiased Contrastive Learning'', NeurIPS 2020.
>
> [h] J. Zbontar, et al., ''Barlow Twins: Self-Supervised Learning via Redundancy Reduction'', ICML 2021.
>
> [i] Y. Tian, et al., ''Understanding Self-Supervised Learning Dynamics without Contrastive Pairs'', ICML 2021.

---

> > ### Comment · Reviewer_wYZX · 2021-08-31
> > **Thanks for the responses**
> >
> > I appreciate the authors' efforts in responding to my original review. Indeed, my review did not focus on any concrete part of this work, but was just concerned with the technical contribution. After reading the rebuttal, I would like to say a few more words.
> >
> > First, I did not say that using strong augmentation contradicts your work -- currently, the SSL methods heavily rely on (moderately strong) augmentations to learn meaningful representation -- I cannot say this is good but it is part of the currently best solution. The proposed method has the potential of solving the issues brought by augmentation, so I would like to see the integration of this work and [A,B] mentioned earlier.
> >
> > Second, it is indeed my personal opinion that the SSL methods based on augmentation (including SimCLR, MoCo, BYOL, etc.) are not essential. They can just learn a simple fact: images generated by these augmentations do not change the semantics, but nothing more. I understand that this paper is based on the current SSL framework and it is unfair to judge the paper based on a to-be-solved problem of the community. However, I would just say that the contribution based on the current framework is limited to me, and this paper seems a bit below the NeurIPS standard.
> >
> > If there is a rating of 5.5, I would like to choose it. Though I will not be upset if this paper gets accepted (I would deliver this message to the AC), it is also difficult for me to give a rating of 6.

---

> > > ### Author Response · Authors · 2021-09-01
> > > **Thanks for your feedback.**
> > >
> > > Many thanks for your reply. We feel sorry that our previous response did not completely clear up your concerns. While we might not be able to change your position at this time, we still find it necessary to share our thoughts as final remarks.
> > >
> > > Firstly, your suggested UOTA+[a] formulation has been shown effective as reported in the earlier response, showing UOTA's robustness against more baselines. We hope you are reassured at least on this point.
> > >
> > > Secondly, we believe that UOTA already delivers non-trivial technical result that would effectively raise a warning flag to the community, and UOTA might be of interest to many audience. Indeed, as you mentioned, augmentations turn out to be the bread and butter for the most successful state-of-the-art SSL approaches. But right owing to this reason, we consider it to be the urging motivation why we were attempting to probe the hidden pitfalls behind such prevailing augmentation pipelines in this paper. We believe we are the first group to examine the variance-bias trade-off existing behind such SSL augmentation pipelines, and we have also gained useful insights into what happened to the pre-trained network parameters if such trade-off failed to be properly balanced. Through rigorous analysis, we find importance sampling can elegantly help reduce the estimator error through careful error decomposition. UOTA then is proposed, in order to justify both theoretically and empirically the correctiveness of our assumption, supporting our discovery of such hidden trade-off behind SSL approaches. We believe UOTA encourages rethinking the structured drawbacks of existing augmentation pipelines for mainstream SSL approaches from a purely statistical perspective, and we certainly look forward to embracing better solution beyond UOTA in this regard.
> > >
> > >
> > > Finally, your opinion that mainstream ''SSL methods based on augmentation (including SimCLR, MoCo, BYOL, etc.) are not essential'' and ''a simple fact that augmentations do not change the semantics, but nothing more'' is worthy of deeper discussions. However, given the empirical success of current SSL (based on augmentation) frameworks, more such SSL architectures and frameworks are proposed, examining and pushing the envelope from an empirical side [j] [k] [l]. Accordingly, there are many interesting and exciting theoretical work emerging, well explaining the provable principle behind success beyond the seemingly ''semantic consistency'' [m] [n] [o] [p] and shows the potential to improve it [q], based on the existing augmentation pipeline. We think both of these empirical study and theoretical works are helping the SSL community progress in a healthy way, in order to figure out a better solution. We believe UOTA is one of such attempts, too.
> > >
> > > Again, we feel grateful for your kind comments and sharing your opinion with us, particularly at this challenging COVID time.
> > >
> > > [j] X. Chen, K. He. ''Exploring Simple Siamese Representation Learning'', CVPR 2021.
> > >
> > > [k] J. Zbontar, et al., ''Barlow Twins: Self-Supervised Learning via Redundancy Reduction'', ICML 2021.
> > >
> > > [l] X. Chen, et al., ''An Empirical Study of Training Self-Supervised Vision Transformers'', arXiv 2021.
> > >
> > > [m] S. Arora, et al, ``A Theoretical Analysis of Contrastive Unsupervised Representation Learning'', ICML 2019.
> > >
> > > [n] Y. Tian, et al., ''Understanding Self-Supervised Learning Dynamics without Contrastive Pairs'', ICML 2021.
> > >
> > > [o] G. Roeder, et al, ''On Linear Identifiability of Learned Representations'', ICML 2021.
> > >
> > > [p] J. HaoChen, et al., ''Provable Guarantees for Self-Supervised Deep Learning with Spectral Contrastive Loss'', arXiv 2021.
> > >
> > > [q] J. Song, et al., "Multi-label Contrastive Predictive Coding." NeurIPS 2020.

---

> ### Author Response · Authors · 2021-08-31
> **We look forward to your feedback.**
>
> Dear Reviewer wYZX, it seems we have not received your feedback on our response yet. As the discussion period will end soon, we were wondering if there are anything else we could address to help further clarify? We sincerely look forward to your feedback. Thanks again for your time!

---

### Official Review · Reviewer_MNMm · 2021-07-20

**Rating:** 6
**Confidence:** 2

**Summary:**

The paper is generally concerned with self-supervised contrastive learning, such as SimCLR. The authors posit that existing ways to generate (contrastive) views introduce out-of-distribution examples that actually harm performance. To remedy this, the authors propose a latent variable model they term UOTA that, based on importance sampling, suggest good view candidates. They show UOTA's theoretical properties and conduct extensive experiments, showing gains using UOTA against a large number of recent baselines.

**Ethical Concerns:**

No ethical concerns.

**Limitations And Societal Impact:**

Yes

**Main Review:**

The motivation for the paper is good -- contrastive learning is a hot topic recently, and as far as I'm aware there is little work describing the harm of using existing view generation / data augmentation strategies. The paper is a bit hard to follow -- the flow and general exposition can be improved, and presenting UOTA as an algorithm block could improve readability. I felt the experimental evaluation was solid -- testing using MoCo, BYOL, and SwAV, with Focal loss, Generalized Cross Entropy, MIL-NCE, Debiased, etc. I felt the ablations were complete as well. The strong and complete empirical results push me to lean in favor of acceptance, however I encourage the authors to improve the flow of the first half of the paper. I did not thoroughly check the proofs in the Appendix, and I'm not too confident about the related works on this particular topic.

**Time Spent Reviewing:**

1

---

> ### Author Response · Authors · 2021-08-09
> **Response to Reviewer MNMm**
>
> Thanks for acknowledging our key contributions. We sincerely appreciate your comments and we hope this response letter can address your concerns.
>
> **A.1.** Firstly, we apologize for the presentation issues. Surely, we would carefully polish throughout the paper to achieve a better organization and presentation quality. Particularly, we will improve the story flow in order to reduce the technical ambiguity and to improve readability. We will make sure all technical details are properly associated with clear explanations and references.
>
> **A.2.** We did not include an algorithm flow in the main paper due to space limit, but we have already included an algorithm flow in the supplementary material. We hope that flow block in the supplementary file may help. If space limits allows, we are definitely happy to move the algorithm flow back into the main paper in an updated version. Thanks again for your comments!

---

### Decision · Program_Chairs · 2021-09-27

**Decision:**

Accept (Poster)

**Comment:**

There was a robust discussion amongst the reviewers about the merits of this work. There was some disagreement as to whether this work's potential impact on the field of semi-supervised learning is big enough. I decided to side with the reviewers that argued that this contribution could in fact be important. The reviewers have unanimously suggested that the new experimental results be included in the final version, as well improved flow & readability.